# Tagging active neurons by soma-targeted Cal-Light

Jung Ho Hyun [1,2,10,11], Kenichiro Nagahama[1,11], Ho Namkung[3,11], Neymi Mignocchi[2], Seung-Eon Roh[1], Patrick Hannan[1,2], Sarah Krüssel [1,2], Chuljung Kwak[1], Abigail McElroy[1], Bian Liu [1], Mingguang Cui [4,5], Seunghwan Lee [4,5], Dongmin Lee [4,5], Richard L. Huganir [1], Paul F. Worley[1], Akira Sawa [1,3,6,7,8,9] & Hyung-Bae Kwon [1,2,3] ✉

Verifying causal effects of neural circuits is essential for proving a direct circuit-behavior relationship. However, techniques for tagging only active neurons with high spatiotemporal precision remain at the beginning stages. Here we develop the soma-targeted Cal-Light (ST-Cal-Light) which selectively converts somatic calcium rise triggered by action potentials into gene expression. Such modification simultaneously increases the signal-to-noise ratio of reporter gene expression and reduces the light requirement for successful labeling. Because of the enhanced efficacy, the ST-Cal-Light enables the tagging of functionally engaged neurons in various forms of behaviors, including context-dependent fear conditioning, lever-pressing choice behavior, and social interaction behaviors. We also target kainic acid-sensitive neuronal populations in the hippocampus which subsequently suppress seizure symptoms, suggesting ST-Cal-Light's applicability in controlling disease-related neurons. Furthermore, the generation of a conditional ST-Cal-Light knock-in mouse provides an opportunity to tag active neurons in a region- or cell-type specific manner via crossing with other Cre-driver lines. Thus, the versatile ST-Cal-Light system links somatic action potentials to behaviors with high temporal precision, and ultimately allows functional circuit dissection at a single cell resolution.

Selective labeling and manipulation of behaviorally-engaged neuronal populations are critical for verifying their causal functions. To target active neuronal populations, immediate early gene (IEG)-based tagging systems have been widely used[1–5]. However, IEGs and their derivatives have limited behavioral applicability because of their poor temporal resolution (several hours) and weak coupling to cell firing[6]. To overcome these limitations, recently developed techniques implemented a dual light- and calcium-dependent switch system, allowing investigators to label active cells with higher temporal precision[7–10]. In brief, these systems used "AND gate" logic, such that gene expression is

[1]Solomon H. Snyder Department of Neuroscience, Johns Hopkins School of Medicine, Baltimore, MD 21205, USA. [2]Max Planck Florida Institute for Neuroscience, Jupiter, FL 33458, USA. [3]Department of Biomedical Engineering, Johns Hopkins School of Medicine, Baltimore, MD 21205, USA. [4]Department of Anatomy, Korea University College of Medicine, Seoul, Republic of Korea. [5]BK21 Graduate Program, Department of Biomedical Sciences, Korea University College of Medicine, Seoul, Republic of Korea. [6]Department of Psychiatry, Johns Hopkins School of Medicine, Baltimore, MD 21205, USA. [7]Department of Genetic Medicine, Johns Hopkins School of Medicine, Baltimore, MD 21205, USA. [8]Department of Pharmacology, Johns Hopkins School of Medicine, Baltimore, MD 21205, USA. [9]Department of Mental Health, Johns Hopkins Bloomberg School of Public Health, Baltimore, MD 21025, USA. [10]Present address: Department of Brain Sciences, DGIST, Daegu, Republic of Korea. [11]These authors contributed equally: Jung Ho Hyun, Kenichiro Nagahama, Ho Namkung ✉e-mail: hkwon29@jhmi.edu

initiated when both Ca²⁺ and light are present. Because action potentials cause, the opening of voltage-dependent Ca²⁺ channels followed by Ca²⁺ influx, light illumination during a specific time period enables gene expression only in neurons active during the behavior.

We previously reported that the Cal-Light technique was useful for specific labeling of the neuronal populations engaged with a bout of behavior during a lever-pressing task in mice: inhibiting the activity of the labeled neurons resulted in behavioral impairment[8]. Labeling and controlling an irrelevant population of neurons that are not associated with lever pressing did not cause deficits in lever pressing behavior, indicating Cal-Light's high selectivity[8]. Recently, the Cal-Light technique was proven to be useful for exclusively tagging and manipulating higher-order cognitive behaviors, demonstrating its broad applicability[11].

Cal-Light system requires two separate synthetic proteins containing C- and N-terminus of tobacco etch virus (TEV), which are fused to calmodulin (CaM) and M13 protein, respectively. When cytosolic Ca²⁺ level increases, CaM and M13 proteins bind to each other, causing the restoration of TEV protease functions. Upon blue light illumination, the TEV cleavage sequence (TEVseq) is exposed to the cytosol, recognized by TEV protease, and allows tethered tetracycline-controlled transcription activator (tTA) to go to the nucleus initiating gene expression. Despite such an advanced design, some caveats still remain. One major weakness could be a portion of gene expression caused by spontaneously occurring background Ca²⁺ signals. Such transient Ca²⁺ influxes arising through various routes would result in the accumulation of action potential-independent labeling. For instance, Ca²⁺ concentration increases are caused not only by somatic action potentials but also by local dendritic Ca²⁺ spikes or synaptic activity in dendrites or dendritic spines[12–14]. Internal Ca²⁺ stores are another source of Ca²⁺ transients[15]. These local sources of Ca²⁺ rise are generally driven by incoming inputs and do not always result in action potential outputs. Especially in awake-behaving animals, neurons are spontaneously active in most of the brain areas, although they are not related to behaviors. These intrinsically occurring background signals will cause Ca²⁺ rise at a number of synapses and dendritic branches, eventually causing aggregation of gene expression at the cell body. Particularly when Cal-Light is applied in vivo using viral infection, which requires several weeks for sufficient expression of Cal-Light constructs, non-specific signals may accumulate over time, resulting in decreased specificity.

To resolve these weaknesses, we develop a soma-targeted version of Cal-Light (ST-Cal-Light). By concentrating the expression of Cal-Light constructs in the cell body, the system mainly converts Ca²⁺ signals caused by somatic action potentials into gene expression by reducing the portion of other Ca2+ sources-dependent gene expressions. Thus, cell labeling becomes more dependent on action potential numbers at the soma. Furthermore, because a high amount of Cal-Light expression is condensed in the cell body, its responsiveness to light and Ca²⁺ becomes higher, decreasing the labeling time. The reduction in time thus increases the specificity of successful labeling, making the Cal-Light technique applicable to a much broader spectrum of behaviors. The previous Cal-Light also requires the injection of three viruses. Because the virus infection rate is not 100%, a mixture of three viruses causes expression variation from cell to cell. To reduce this limitation, here we generate the ST-Cal-Light knock-in (KI) mouse line. Furthermore, because the KI mouse is designed to be conditional, breeding with other Cre-driver lines enables the targeting of functionally active neurons in a designated cell type.

## Results

### Development and characterization of ST-Cal-Light in vitro

We used two different soma-targeting peptides to facilitate the localization of Cal-Light to the cell body membrane. The first was a 150 amino acids fragment at the amino terminus of kainate receptor 2

(KA2)[16], and the other was a 65 amino acids fragment at the carboxyl terminus of the voltage-dependent potassium channel, Kv2.1[16,17]. To test whether these two motifs restrict Cal-Light expression to the cell body, we inserted the soma-targeting fragment between the cytosolic side of the transmembrane (TM) domain and calmodulin (CaM) of the main Cal-Light construct (Myc-TM-*KA2 or Kv2.1 motif*-CaM-TEV-N-AsLOV2-TEVseq-tTA) (Fig. 1a). Another Cal-Light component, M13-TEV-C, is allowed to freely diffuse in the cytosol without having any localization motif so that it can interact with the other partner without spatial restriction.

To confirm somatic localization, we performed antibody staining of myc tag, an epitope added at the outer membrane side of the TM domain. We first examined the localization of the original Cal-Light (OG-Cal), which was found to be expressed in both cell body and dendritic branches (Fig. 1b). Addition of the soma-targeting Kv2.1 or KA2 motif, named ST-Kv2.1 and ST-KA2, respectively, caused a rapid signal reduction in dendrites, indicating preferred localization at the cell body (Fig. 1b, c). Cell-fill fluorescence (tdTomato) was broadly distributed throughout all processes of neurons. To test whether gene expression is dependent on Ca²⁺ and light, we transfected either OG-Cal, ST-Kv2.1, or ST-KA2 with a TetO-EGFP reporter into hippocampal culture neurons. Na⁺ channel blocker, tetrodotoxin (TTX), was applied to create a no-activity condition, whereas GABA_A receptor antagonist, bicuculline, was applied to increase overall neuronal activity. In a dark condition, increasing neuronal activity by bicuculline did not increase EGFP reporter gene expression, and TTX also did not further reduce gene expression level (Fig. 1d–g). These results confirmed that neuronal activity alone was not sufficient to induce gene expression if the light is absent, but both blue light (473 nm) and bicuculline-induced gene expression robustly (Fig. 1d–g).

All three Cal-Light constructs showed robust Ca²⁺- and light-dependent reporter gene expression, but a higher signal-to-noise ratio (SNR) was observed in the case of ST-Kv2.1 and ST-KA2 (1.2-fold higher for ST-Kv2.1 compared to OG-Cal, $p < 0.01$; 1.8 fold higher for ST-KA2 compared to OG-Cal, $p < 0.005$) (Fig. 1g). To examine the pattern of gene expression from individual cells, a scatter plot of red and green fluorescence was analyzed (Fig. 1h and Supplementary Fig. 1). Exceptionally strong green signals were found in a few neurons with ST-Kv2.1 infection despite weak red fluorescence. This mismatched reporter gene expression was very low in ST-KA2 transfected cells. Thus, ST-KA2 resulted in the highest SNR.

To verify whether similar results are also obtained in slice cultures, we infected adeno-associated viruses (AAV) expressing either ST-Kv2.1 or ST-KA2 into organotypic cortical slice cultures. A stimulation pipette was localized at layer 2/3, and a train of action potentials (5 pulses at 10 Hz per minute) with 10-s long blue light was delivered for 15 min. Similar to experiments in dissociated cultures, robust gene expression was induced when both electrical stimulation and blue light were present simultaneously, but not by either individually (Fig. 1i). The ratio of green to red fluorescence was significantly higher in ST-Kv2.1 and ST-KA2 compared to the OG-Cal (1.7-fold higher for ST-Kv2.1 compared to OG-Cal, $p < 0.005$; 2-fold higher for ST-KA2 compared to OG-Cal, $p < 0.005$) (Fig. 1j). The overall fluorescence plot revealed that ST-KA2 resulted in almost no background EGFP reporter gene expression, but a small amount of light-independent expression (resulting from activity only) was found in neurons expressing ST-Kv2.1 (Fig. 1k). Because of lower background signals and high induction capability, we decided to further test ST-KA2 constructs in behaving animals.

### Labeling and manipulation of active neurons underlying lever-pressing behavior

To determine whether ST-KA2 can selectively label behaviorally specific neuronal populations, we trained mice for a lever-pressing task as described previously[8] (see Methods). AAV-ST-KA2 was bilaterally

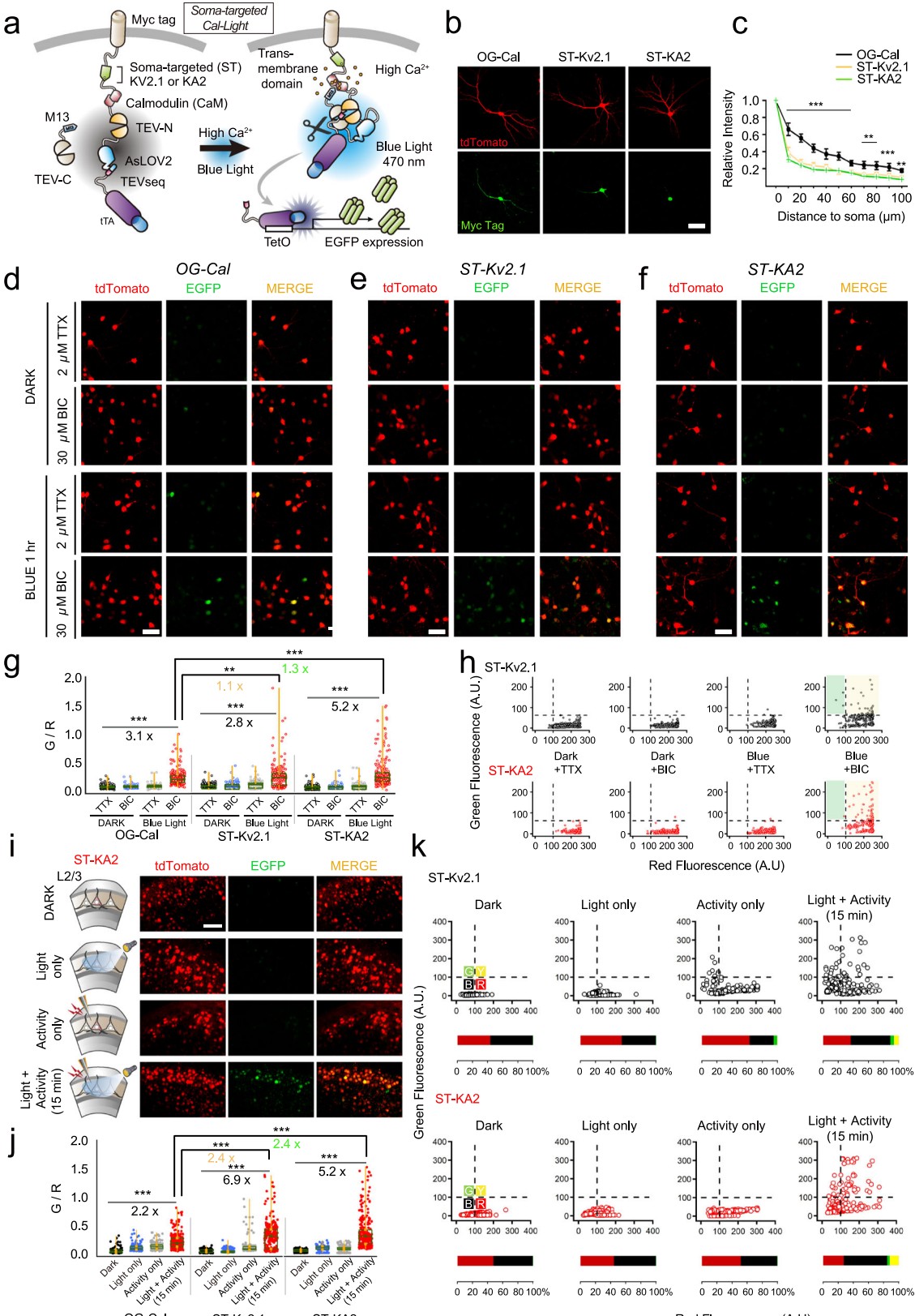

injected into layer 2/3 of the primary motor cortex (M1) (Fig. 2a). Briefly put, water-restricted mice underwent continuous reinforcement (CRF), during which the mice learn lever pressing is associated with water rewards (Fig. 2b). The next sessions are fixed ratio (FR) training with a gradual increase in the number of lever presses required to receive a reward. For labeling, we shone blue light (5 s) whenever the mouse pressed a lever, but then the blue light was prohibited for 25 s, even if the mouse presses a lever continuously (5 s ON/25 s OFF cycle) (Fig. 2c). To obtain the maximum level of labeling, labeling protocol started from CRF session 15 (CRF15) to FR session 12 (FR12) (Full label). To test whether ST-KA2 enables efficient labeling with shorter period of blue light illumination time, we also tested a mild labeling process in

**Fig. 1 | Development and verification of soma-targeted Cal-Light. a** Graphical illustration of soma-targeted Cal-Light system. Elevation of $Ca^{2+}$ concentration in the cytosol causes M13 and CaM protein interaction which causes binding of c- and n-terminus of tobacco etch virus (TEV) protease (TEV-C and TEV-N). When TEV-C and -N fragment bind, they regain proteolytic functions; however, the TEV recognition sequence (TEVseq) is buried inside of AsLOV2 Jα-helix, so TEV protease access to the TEVseq is prohibited. Blue light triggers structural changes of AsLOV2, rendering the TEVseq exposed to the cytosol. Then, TEV protease cleaves out tTA, and gene expression begins. **b** Confocal images of the cell transfection marker, tdTomato (red), and antibody staining of myc epitope (green). The scale bar indicates 50 μm. **c** The degree of expression in soma and dendrites. The green-to-red signal (G/R) ratio was normalized to the cell body, and the ratio was measured from the cell body up to 100 μm in dendrites. **d**–**f** Representative images of EGFP reporter gene expression in various conditions. Original Cal-Light (**d**), ST-KV2.1 (**e**), and ST-KA2 constructs (**f**) were transfected in hippocampal culture neurons, and neural activity was controlled by 2 μM TTX or 30 μM bicuculline (Bic). The scale bar indicates 50 μm. **g** Comparison of gene expression. G/R ratios from individual cells are plotted. An intermittent flash of blue light (1 s ON/9 s OFF) was illuminated for 1 h G/R and was analyzed with one-way ANOVA. Asterisks (**$p < 0.01$ and ***$p < 0.001$) indicate Bonferroni post hoc significance. **h** Scatter plot of G/R. Open circles indicate individual neurons. Yellow-colored area indicates neuronal population with both green and red signals. The green area indicates cells with a green fluorescence alone. **i** Schematic of experimental conditions. Representative images from each condition (dark, light only, activity only, light + activity) are shown. tdTomato is a transfection marker, and EGFP is a reporter. Scale bars, 80 μm. **j** Summary graph of the G/R ratio from cells transfected with OG-Cal, ST-Kv2.1, and ST-KA2, respectively. G/R values from individual neurons and a summary box plot chart are superimposed. The magnitude was robustly enhanced when both light and activity were present. **k** Green and red fluorescence values from individual neurons were plotted. Individual neurons were divided into four groups (green, yellow, red, and black) by the level of red and green fluorescence. The horizontal bar graph represents the percentage of red, black, green, and yellow groups. For all graphs, *,**, and *** indicate $p < 0.05$, $p < 0.001$, and $p < 0.005$, respectively. Box plots show the median, 25th and 75th percentiles, and whiskers show min to max. Error bars indicate s.e.m. Source data and statistics are provided in the Source Data file.

which blue light was illuminated only during FR 8–12 (Fig. 2b). The total blue light exposure time was about 550 s for full labeling and 350 s for mild labeling (Fig. 2d, g). These protocols, especially for full labeling, were performed in order to match the same conditions in which the previously developed Cal-Light technique was tested[8]. In both cases, gene expression was strongly induced, but the full labeling condition showed more robust gene expression (Fig. 2e, f). As exemplified in Fig. 2e, major changes were only green signals in "Light + Activity" conditions. R signals were similar across all conditions. To avoid any thresholding effect set by the R signal, we analyzed only green signals. As expected, the results were similar to those shown in G/R (Supplementary Fig. 2). We further examined single session (FR12) labeling, which resulted in the significant increase of gene expression (Fig. 2f).

To determine whether altering the activity of the labeled neurons is sufficient to perturb the learned lever-pressing behavior, we injected ST-KA2 viruses together with a halorhodopsin (NpHR) reporter (Fig. 2h). Two days after the labeling procedure, we delivered 589 nm light to suppress the activity of the labeled neurons by activating NpHR (Fig. 2i). The number of lever presses was counted as an indication of learned behavior (Fig. 2j). Once mice finished FR12 training, 589 nm light (2 s ON/1 s OFF) significantly reduced the lever pressing number regardless of full or mild labeling (Fig. 2j). These results indicate that labeling with a shorter period of blue light illumination (~ a few minutes) was enough to prevent learned behavior. Normal lever pressing behavior was fully recovered on the following day, suggesting that the perturbed behavior was not due to tissue damage or long-term circuit changes by the yellow laser (Fig. 2j, k). Consistent with the reduced number of lever presses, a longer time was needed to reach 250 lever presses, and inter-reward intervals were prolonged in the presence of 589 nm light (Fig. 2l, m). Even if mice pressed the lever, a significant portion of lever pressing events was not coupled to water rewards, reflecting the inhibition of recalling learned memory (Fig. 2n). When an EGFP reporter was used as a negative control, the 589 nm light did not impair lever-pressing behavior, verifying that behavioral suppression was indeed caused by NpHR activation, not by the yellow light itself (Fig. 2n).

To measure the degree of off-target expression of NpHR without light or by 589 nm light, we performed additional experiments. When blue light was not delivered or yellow light (589 nm) was used for labeling, no EGFP reporter was expressed (Supplementary Fig. 3). To test the efficacy of cell labeling depending on the distance from the light source, we tested two different lengths of optic fibers. In one case, the tip of the optic fiber was positioned 0.4 mm above the virus injection site, and in the other case, the optic fiber was located 0.8 mm above the virus injection site. The blue light was delivered to the mPFC for 30 min (2 s ON/1 s OFF) in both conditions. Interestingly, we found that the labeling efficiencies were similar in terms of G/myc ratio.

Robust EGFP reporter gene expression was induced even if the optic fiber was positioned 0.8 mm above the virus injection site. These results suggest that the ST-Cal-Light is sufficient to label neurons with various distances from the blue light sources in vivo.

## Application of the ST-Cal-Light in short-lived behaviors

Lowering background noise signals while maintaining high inducibility of gene expression may allow for cell labeling during a short period of time. This is critical for in vivo application because many behaviors, such as social interaction, are transient and not recurrent. To test if ST-KA2 can be applied to such short-lived behaviors, we assessed several types of behaviors. We injected a mixture of viruses expressing ST-KA2, M13-TEV-C, and TetO-NpHR-EYFP into the dorsal hippocampus (Fig. 3a). The neuronal population engaged with contextual fear memory was labeled by shining blue light concomitantly with a foot shock. This paring was repeated 3 times (1-min interval), so the total duration of blue light exposure was 15 s (Fig. 3a). Two days after fear acquisition, mice displayed freezing responses in the conditioned context during the retrieval phase as anticipated; however, 589 nm light robustly reduced the percentage of freezing (Fig. 3b). We confirmed robust NpHR gene expression in a broad area of the dorsal hippocampal (Fig. 3c–e). Thus, ST-KA2 efficiently labeled cell populations encoding contexture fear memory through minimal bouts of light flashes, and their behavioral causality was verified.

To test if reactivation of the ST-Cal-Light-labeled neurons is sufficient to cause freezing behavior, we performed the same fear conditioning experiment where ChrimsonR was expressed instead of NpHR (Fig. 3f). Blue light was shone for 5 s concomitantly with a foot shock and this labeling was repeated 3 times. During the reactivation session, mice were exposed to a novel context (context B) distinct from the conditioning context (context A). Freezing behavior was triggered upon ChrimsonR stimulation with a 589 nm laser, whereas the control mice did not display freezing behavior despite light stimulation (Fig. 3g).

To further dissect the temporal extent to which the ST-Cal-Light can label active neurons and show behavioral causality, we tested labeling only with a single foot shock (Supplementary Fig. 4a). During contextual retrieval, stimulating NpHR with yellow light (589 nm) did not prevent freezing behavior (Supplementary Fig. 4b, c). We also directly compared the labeling efficiency of the ST-Cal-Light with that of the OG-Cal-Light. When labeled with the ST-Cal-Light, three context-shock pairings were sufficient to demonstrate behavioral causality. However, the same labeling protocol did not show behavioral changes when the OG-Cal-Light was used (Supplementary Fig. 4d–f). These behavioral differences were consistent with the differences in labeling intensity (Supplementary Fig. 4g).

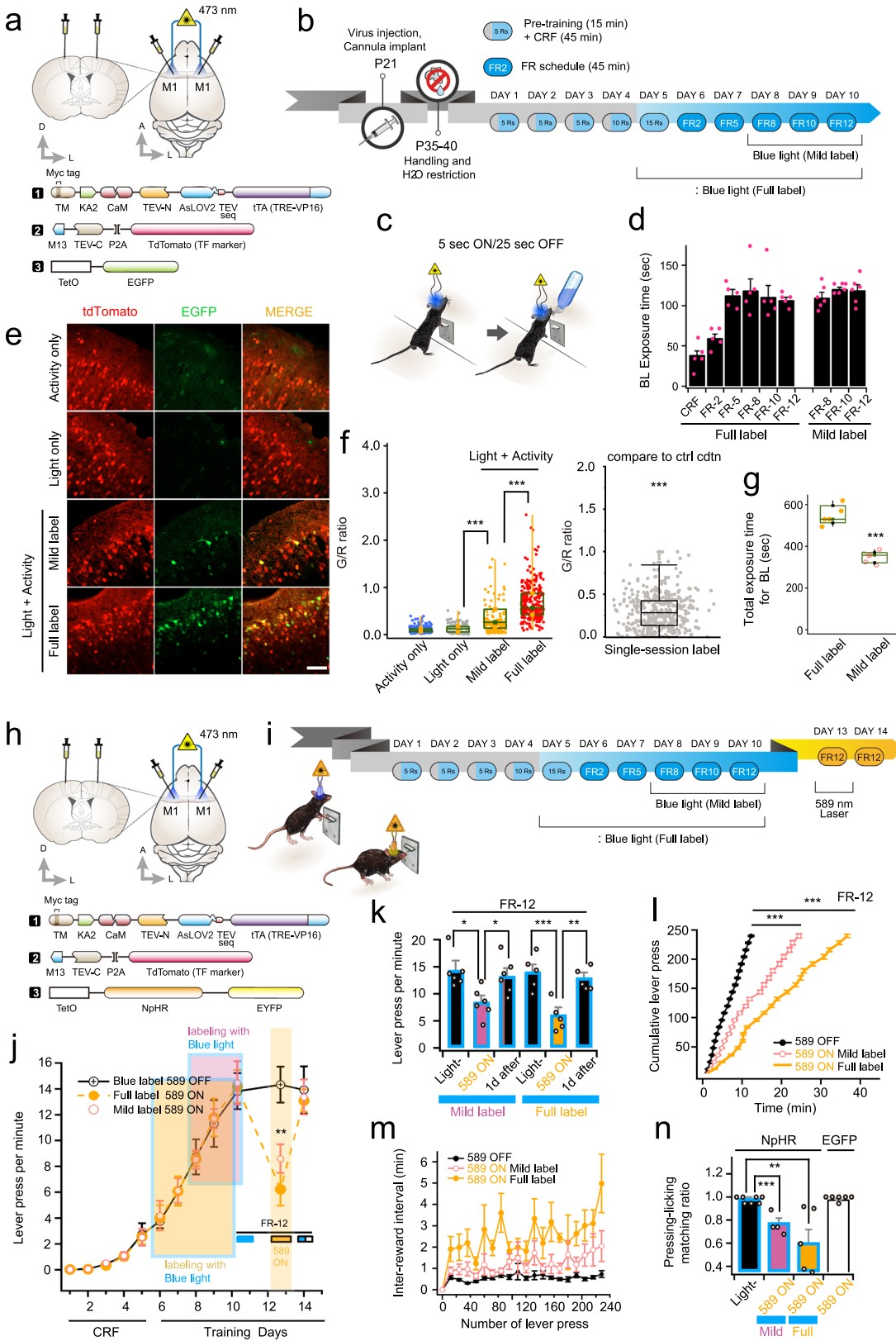

We further tested whether ST-KA2 can also target neurons involved in behaviors based on high cognitive functions. The medial prefrontal cortex (mPFC) has been well characterized for mediating social cognition[18–22], but functionally involved neural populations have not been analyzed at an individual cell resolution. To test whether ST-KA2 can selectively label an mPFC neuronal population engaged with social interaction, a mixture of AAVs expressing ST-KA2, M13-TEV-C,

and TetO-EGFP were bilaterally injected into the layer 5/6 of the mPFC (Fig. 3h). For reporter gene induction, a blue light was programmed to be switched on for 5 s whenever the test mouse entered a social zone (3 cm from the social box) to interact with another mouse (Fig. 3i). Because of the mouse to mouse variability in interaction time, the amount of blue light exposure was different in each animal. *Post-hoc* confocal imaging revealed that EGFP reporter gene expression was

**Fig. 2 | Labeling and control of active neurons engaged with lever-pressing behavior. a** ST-KA2 viruses with AAV-TetO-EGFP were bilaterally injected into layer 2/3 of the primary motor cortex. **b** Schematic mouse training schedule and labeling procedures with blue light. 5Rs, 10Rs, and 15Rs indicate that mice receive five (5Rs), ten (10Rs), and fifteen rewards (15Rs), respectively. **c** Fiber optics for blue light illumination were implanted in both sides of the primary motor cortex. **d** Blue light exposure time per training day was measured. tdTomato signal indicates the efficiency of viral injection. Higher green fluorescence was observed as blue light exposure time increases. Scale bars, 100 μm. **e** When cell labeling is finished, the brain was fixed, and the degree of gene expression was quantified by confocal imaging. **f** Individual G/R ratios with the box plot chart superimposed. **g,** A box plot chart for total blue light exposure time at each condition. **h** Schematic drawing of virus injection and fiber optic implantation (top) and injected three viral constructs (bottom). **i** Mouse training, labeling by blue light, and halorhodopsin inhibition procedures. Active neurons during lever pressing were labeled by blue light, and their activity was inhibited by 589 light. **j** Total number of lever presses increased over training days. Periods of blue light labeling were indicated by shaded boxes

with different colors (five mice for full labeling, and six mice for mild labeling). The number of lever presses was significantly reduced by a 589 nm laser but fully recovered the following day in the absence of 589 nm light (*inset*: blue horizontal bar underneath the FR-12 label indicates the last day of training with labeling in the presence of blue light. The yellow horizontal bar indicates the probe-test day in the presence of yellow light throughout the session (2 s ON, 1 s OFF). The bar half-filled with blue indicates the following day in the absence of yellow light but labeled with blue light during training). **k,** The total lever pressing number was compared before and after 589 nm light, and the following day of the inhibition test. **l** The number of lever presses was plotted over time to demonstrate how fast animals reach the goal. Note that more time was required to reach 250 lever presses when the 589 nm light is turned on after labeling. **m** Inter-reward interval was prolonged when the yellow light was turned on. **n** Summary graph of lever pressing-licking matching ratio. For all graphs, *,**, and *** indicate $p < 0.05$, $p < 0.001$, and $p < 0.005$, respectively. Box-and-whisker plot shows the median, 25th and 75th percentiles, and whiskers show min to max. Error bars indicate s.e.m. Source data and statistics are provided in the Source Data file.

present in a subset of neurons (Fig. 3j). Additionally, EGFP expression was positively correlated with the amount of blue light exposure (Fig. 3k). These results confirmed that the magnitude of gene expression mediated by ST-Cal-Light is dose-dependent. To confirm the behavioral causality of the labeled neurons, we shined a 589 nm light in animals injected with an EGFP reporter for the control group and an NpHR reporter vector for the test group. The control group administered with the yellow light did not cause any changes in social interaction behavior (Fig. 3l), but the same yellow light to the test group expressing the NpHR reporter reduced the degree of social interaction as indicated by a reduced social preference index (Fig. 3m). Thus, the ST-Cal-Light could specifically visualize and confirm the behavioral causality of neurons governing social interaction.

## Application of the ST-Cal-Light in brain diseases

We next tested whether ST-Cal-Light can also be used to dissect neuronal populations underlying specific brain disorders. It is generally believed that epileptic seizure is caused by uncontrolled electrical discharge in the hippocampus[23]. Optogenetic approaches to control epileptic seizure have been demonstrated in the past decade, but the controlling target was a specific brain area or a cell type[24–26]. These approaches affect the activity of the entire target brain area or cell population, which includes both disease-related and normally functioning neurons. Interrogating only seizure-engaged neurons would have benefited by minimizing the side effect but resulting in a similar amelioration of seizure symptoms. A mixture of viruses (AAV-ST-KA2, AAV-M13-TEV-C-P2A-TdTomato, and TetO-EGFP) was injected in the hippocampal dentate gyrus (DG) and CA1 areas (Fig. 4a). A seizure was induced by administrating kainic acid (KA) intraperitoneally (20 mg/kg) (Fig. 4b). Shortly after KA injection (10 min), blue light (3 s ON/2 s OFF, 30 min) was illuminated in order to label active neuronal populations linked to an epileptic seizure. In a control group, the same amount of KA was administered, but blue light was not delivered to determine whether labeling is light-dependent or not. As predicted, robust gene expression was only present in the group which received blue light illumination (Fig. 4c). If labeled neurons are directly coupled to behavioral abnormalities, we may be able to ameliorate seizure symptoms by selectively suppressing the activity of labeled neurons. For this experiment, an AAV expressing the TetO-NpHR reporter was injected together with AAV-ST-KA2, and AAV-M13-TEV-C-P2A-TdTomato (Fig. 4d). After ST-Cal-Light proteins were fully expressed, the seizure-specific neurons were labeled during the first KA injection. A second KA injection was tested either with or without shining 589 nm light (Fig. 4e). The severity of seizure phenotypes was analyzed by division into multiple stages, including immobilization, head nodding, continuous myoclonic jerk, and clonic-tonic convulsions (see Methods). Symptoms were exacerbated over time after systematic KA injection, but the delivery of

589 nm light significantly suppressed the progression of seizure symptoms (Fig. 4f–h). Inhibiting hippocampal neurons involved in context-dependent fear conditioning did not reduce seizures following KA injection (Fig. 4h). No reduction in seizure behavior might be due to the weak labeling made by a short period of blue light exposure during the fear conditioning (5 s, 3 times). To test the same labeling conditions but the behavioral effect of randomly labeled neurons, we performed additional experiments. In this experiment, we injected the same set of viruses that were used for seizure experiments and trained the animal for lever pressing, shown in Fig. 2. We shone blue light, 3 s ON/2 s OFF, for 5 min during the lever press training (from CRF 15Rs to FR12). The total duration of blue light was 30 min, which was the same duration used for the KA-induced seizure experiments. With this labeling procedure, we observed robust labeling in granule cells, CA1, CA3 neurons, and mossy cells in the hippocampus (Supplementary Fig. 5). However, when we shone 589 nm light after the second KA injection, the epileptic seizure score maintained high (Supplementary Fig. 5). These results suggest that seizure symptoms are mediated by specific cell population, but not by nonspecific hippocampal neurons. Post hoc staining showed that granule cells in the DG, mossy cells in the hilus, and CA1 and CA3 hippocampal neurons were labeled (Fig. 4i). Because the ST-Cal-Light is designed to express higher NpHR expression in more active neurons, it is presumed that highly active neurons such as superhub neurons[27,28] were preferentially inhibited in our experiments. The behavioral rescue could be a result of indirect circuit changes. To directly monitor seizure activity in the hippocampus, we now performed the ST-Cal-Light experiments with EEG recording. An electrode for EEG recording was implanted over the hippocampus together with optic fiber. After cell labeling, the second KA injection caused ictal-like electrical discharges in the hippocampus, which was recognized by large magnitudes of EEG signals (Fig. 4j). These large amplitudes were more often detected in the mouse group in the absence of yellow light. Shortly after the KA injection, seizure activity was more frequently detected over time, and similar progressive seizure symptoms were also observed (Fig. 4k). When 589 nm light was shone, the average seizure activity progression, as well as seizure score, were suppressed, verifying again that the labeled neurons are directly implicated in seizure brain activity (Fig. 4l). Thus, the ST-Cal-Light would be useful for dissecting neural circuits engaged in brain disorders at the individual cell or subpopulation level.

## Generation of conditional ST-Cal-Light knock-in mice

Identification of behaviorally relevant neurons within a selective cell type will improve understanding of the specificity of their roles underlying behaviors. To achieve this goal, we generated a conditional ST-Cal-Light KI mouse (Fig. 5). The ST-KA2 gene was inserted into the GtROSA26 locus (Fig. 5b). A LoxP flanked neo cassette was placed upstream of the target gene, so that cre recombinase can remove the

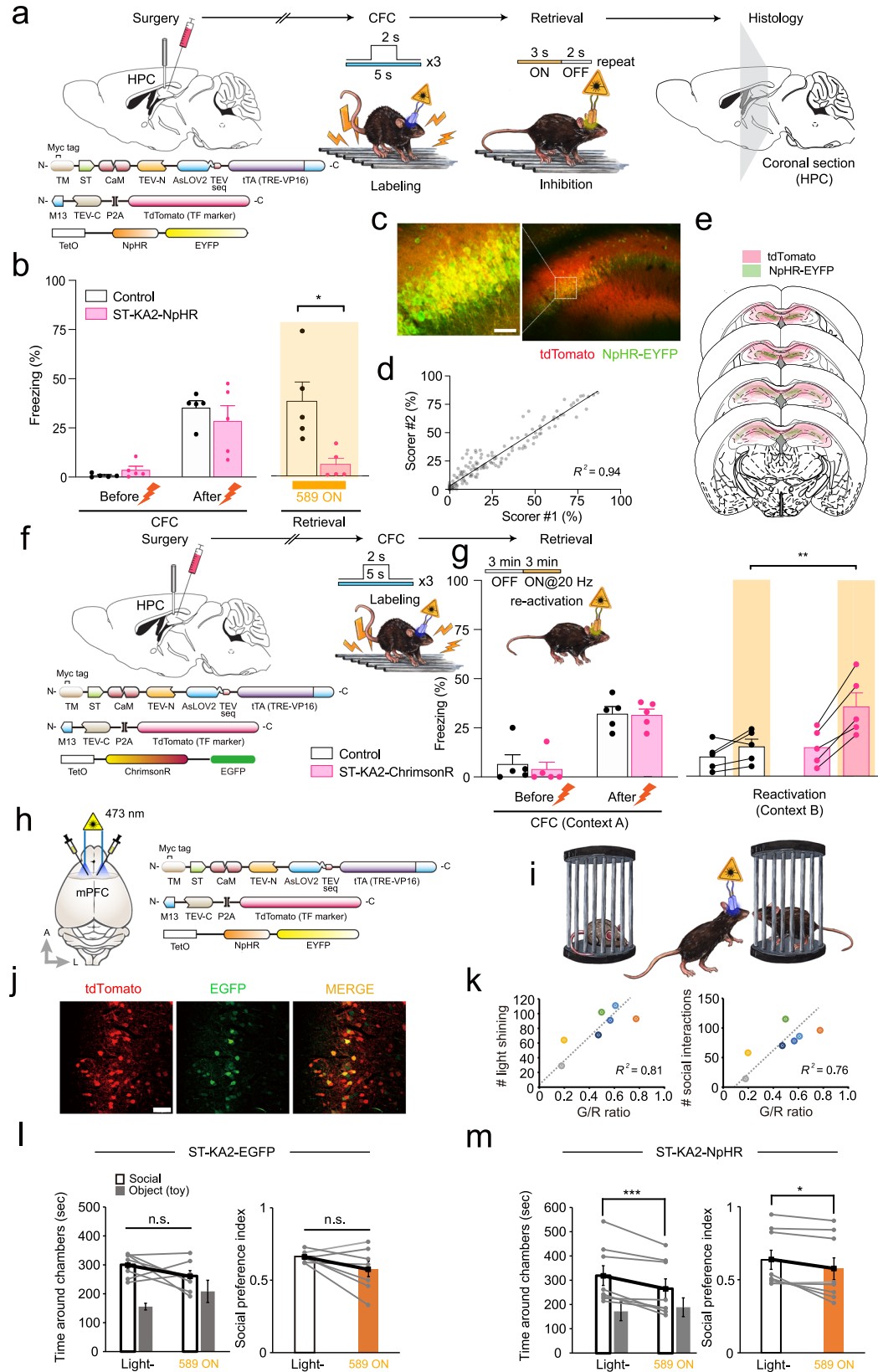

neo cassette, resulting in ST-KA2 expression. To confirm that floxed ST-Cal-Light causes reporter gene expression in a cre-dependent manner, we compared EGFP reporter expression. When AAV expressing cre was not injected into the floxed ST-Cal-Light KI mice, repeated shining of blue light (3 s ON/2 s OFF, 30 min) did not cause EGFP expression (Fig. 5c). In contrast, robust EGFP expression was induced

in mice injected with cre-expressing virus, verifying the KI system works in a cre-dependent manner (Fig. 5c). To make ST-Cal-Light system work in all neurons transfected with M13-TEV-C, we constructed a new viral plasmid expressing both M13-TEV-C and Cre recombinase (Fig. 5d). Introducing this construct with an EGFP reporter showed clear EGFP expression (Fig. 5d). When generating the conditional

**Fig. 3 | Controlling context-dependent fear conditioning and social interaction behavior. a** Schematic illustration of virus injection and fear conditioning experiments. A mixture of viruses expressing ST-KA2, M13-TEV-C, and TetO-NpHR-EYFP was injected into the dorsal hippocampus. Fiber optics were implanted bilaterally above the viral injection site. Short pulses of blue light (5 s × 3 times) were delivered for labeling active neurons, and yellow light was shined during the probe test. Reporter gene expression was confirmed by taking confocal images. **b** The percentage of freezing was compared before and after conditioning, and with or without 589 nm light during the retrieval period. *p < 0.05. Graphs expressed as mean ± SEM. The sample size presents the number of independent mice. **c** Representative image of NpHR-EYFP expression. Scale bars, 50 μm. **d** Freezing score was analyzed by two independent people in a blind manner. The freezing percentage was scored every 10 s and cross-checked with correlation analysis. **e** The extent of virus injection and NpHR-EYFP expression is plotted across several coronal sections of the brain. Images illustrate a series of coronal schematics showing the extent of AAV expression (red) and activity-dependent labeling (green). The extents were traced based on fluorescent images taken at low magnification (2.5×) for each animal (n = 5 independent mice). Darkness represents coincidence from different animals. **f** Schematic illustration of the experimental procedure. Viruses expressing ST-KA2, M13-TEV-C, and TetO-ChrimsonR-EGFP

were injected into the dorsal hippocampus CA1 area bilaterally. Short pulses of blue light (5 s × 3 times) were delivered with a 1-min interval for labeling, and 589 nm yellow light was shined for testing behavioral causality. **g** The percentage of freezing was compared before and after conditioning. During the reactivation session, freezing behavior in a novel context (context B) was compared before and after the delivery of 589 nm light. Reactivation of ST-Cal-Light-labeled neurons was sufficient to trigger freezing in context B. **h** Virus injection and fiber optic implantation scheme. Viruses were injected into layer 5/6 of the mPFC. **i** Cartoon for social interaction experiments. Whenever the mouse entered a social zone, a blue laser connected to the fiber optics was switched on. **j** Sample images of active neuron labeling in the mPFC by ST-Cal-Light. Scale bars, 100 μm. **k** Graphical demonstration of positive correlation between the number of blue light illumination/social interactions and G/R ratio. Each data point corresponds to one mouse. **l** When active neurons were labeled with EGFP reporter, 589 nm light did not decrease social interaction. Data are presented as mean values ± SEM. Statistical significance is judged by a two-tailed paired t-test. **m** NpHR reporter gene was expressed during the labeling process, social interaction behavior was significantly inhibited during the probe test. Source data and statistics are provided in the Source Data file.

floxed ST-Cal-Light KI mouse, a myc epitope was added at the outside of TM domain, so antibody staining against myc showed signals in the outer cell membrane (Fig. 5d).

To test cell type-specific labeling, we crossed the floxed ST-Cal-Light mouse with Emx1-Cre. We injected AAV-M13-TEV-C-P2A-TdTomato and AAV-TetO-EGFP into layer 2/3 of the primary motor cortex of either ST-Cal-Light hetero- or homozygotes (Fig. 5e). tdTomato signal was driven by the CAG promoter, so its expression was not restricted to specific cell types, but EGFP expression was induced in excitatory neurons because Cre expression was limited to neocortical excitatory neurons under the Emx1 promoter[29] (Fig. 5f). Antibody staining with CaMKII, an excitatory pyramidal neuron marker, showed colocalization of CaMKII and EGFP signals (Fig. 5g and supplementary Fig. 6a). Similarly, we tested labeling of active interneurons by crossing the floxed ST-Cal-Light mouse with a PV-Cre mouse (Fig. 5h, i). Repetitive blue light (3 s ON/2 s OFF, 30 min) induced EGFP reporter gene expression restricted to PV interneurons (Fig. 5i and Supplementary Fig. 6b). About half of neurons out of CaMKII- or PV-positive neurons were labeled as indicated by positive EGFP signals. The majority of EGFP-positive neurons (86% and 82%) were also positive to CaMKII or PV antibody staining (Supplementary Fig. 6). Thus, conditional ST-Cal-Light mice enable targeting active neurons out of the genetically defined cell type.

## Discussion

Labeling and manipulating active neurons with the high temporal resolution is crucial for understanding circuit functions underlying animal behaviors. A dual switch system with "AND" gate logic using Ca²⁺ and light has successfully converted fast Ca²⁺ transients to slow gene expression, allowing for activity control of labeled neurons[7–10]. Despite these advantages, collecting all signals arising from various Ca²⁺ sources causes labeling that may not be directly associated with specific actions. To minimize such noise signals, we intended to condense the Cal-Light system, restricting it to the cell body. Highly concentrated Cal-Light proteins at the cell soma increase the chance of binding two Cal-Light components, which results in efficient tTA release upon blue light. Thus, soma-targeted Cal-Light results in higher expression of reporter proteins while simultaneously reducing off-target gene expression. In this scheme, the magnitude of gene expression remains high, but background signals are reduced. Furthermore, because somatic action potentials are always the final output signals of neurons, their inhibition negates integrative synaptic potentials resulting in prominent effects on behaviors. All these improvements collectively make the labeling process more efficient as well as shorten the duration and repetition of blue light illumination.

The optimal condition for the best labeling would be different from the case by case. Dependence on Ca²⁺ and light is the basic operating system of the ST-Cal-Light. What it means is that any factor that affects Ca²⁺ levels and how it is matched to the blue light protocol will be critical. Because different types of cells have different intrinsic properties (e.g., resting membrane potentials, ion channel distributions, input resistance), it is difficult to make a guideline that universally applies to all cells. Nevertheless, several important rules should be considered.

First, the expression level of two ST-Cal-Light components is important. If they are expressed in a similar amount (e.g., 1:1 ratio), the labeling efficiency increases. If the M13-containing construct is much higher or lower compared to the CaM-containing construct, then labeling efficiency decreases[8]. That is because these two constructs work together in order to begin gene expression. Second, labeling efficiency increases if cells fire as a short burst with a high frequency. Even if the number of action potentials and blue light exposure time is the same, the higher reporter gene expression is made if firing occurs as a short burst[8]. Third, the number of blue light repeats is more important than the total duration of blue light. The logic is that the onset timing of AsLOV structural modification by blue light is very fast, but the restoration to the original structure takes tens of seconds. Due to such light responsiveness, a brief light pulse (1–2 s) followed by some interval (~30 s), and then repeating the same protocol will maximize the labeling efficiency.

In this study, we examined the efficiency of ST-Cal-Light in four different behaviors. The first was a lever-pressing behavior, in which animals press a lever repetitively in order to acquire rewards. Tagging active neurons during such repetitive behavior is one good case for using the ST-Cal-Light technique. Because the strength of labeling positively correlates with repeated light exposure and the amount of Ca²⁺ release, more active neurons accrue greater gene expression with each behavioral repeat, while less active neurons do not. Labeling efficiency is maximized if repeated behavior has an inter-trial interval because 30–60 s are required after blue light illumination for the structural restoration of the AsLOV2 domain[30].

Labeling sometimes resulted in low tdTomato expression (infection marker) but strong green fluorescence (reporter). This is not likely the nonspecific EGFP expression in the absence of the TEV-C protease. When we transfected the mutant truncated form of the TEV-C component, almost zero reporter gene expression was observed[31]. This result indicated that the other component (tTA containing construct) could not induce gene expression by itself, or even if that is possible, the level is extremely low. Therefore, the partner construct, TEV-C, must be present in order to make substantial light-dependent gene

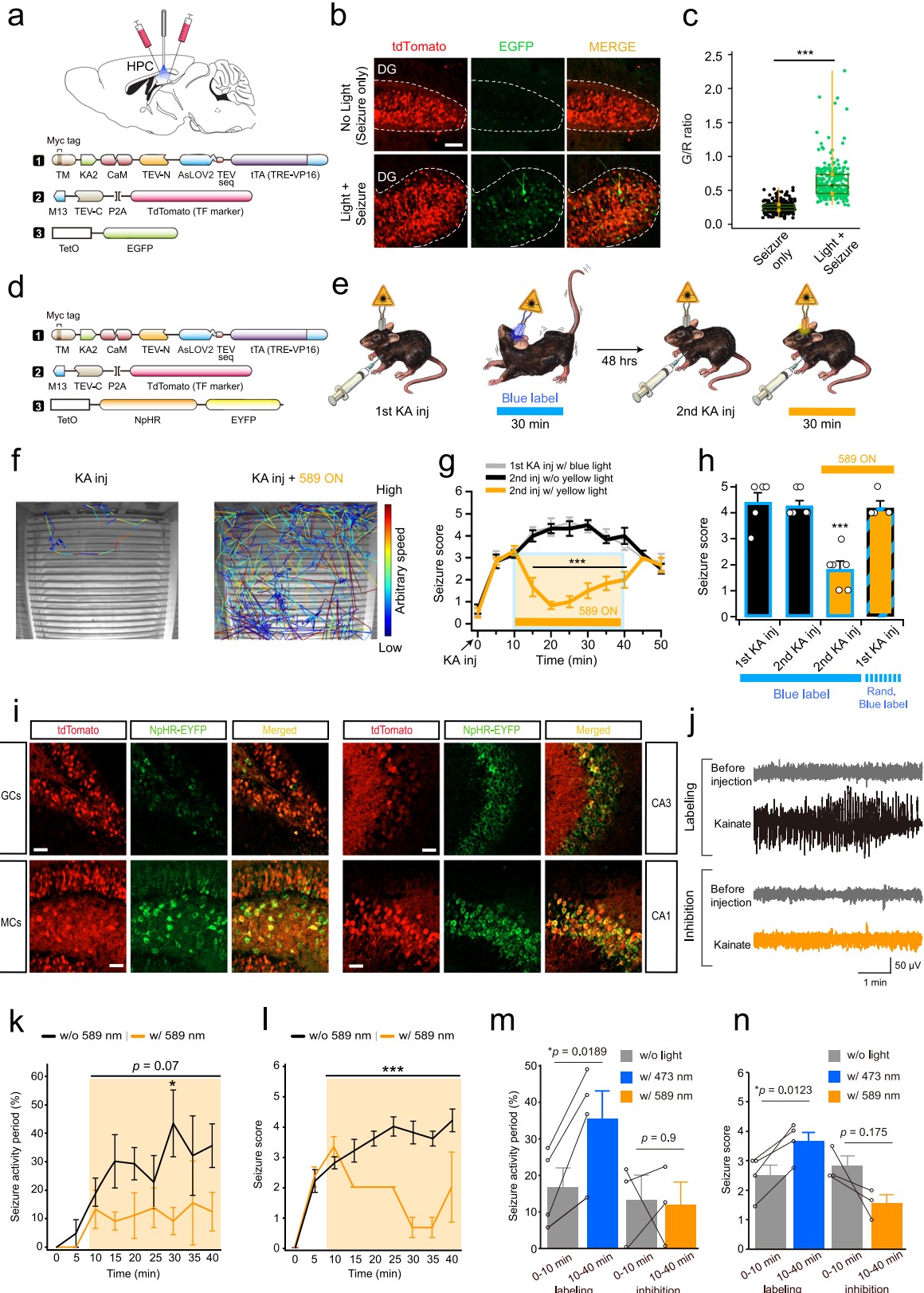

expression. Furthermore, when we expressed TEV-C-P2A-TdTomato, but M13 was deleted (*del*M13-TEV-C-P2A-TdTomato), then almost zero reporter gene expression was made[8,31] (NB). These results again strongly indicate that a very bright EGFP signal was mostly induced in a light and calcium-dependent manner, not by the leaky reporter gene expression nor by adjusted red fluorescence level.

Although we do not know exactly what caused this phenomenon, one possible scenario is that in some neurons, a portion of tTA was cleaved out via normal Cal-Light processes (TEV-C and TEV-N reconstitution and TEV sequence exposure by light), but gene expression became abnormally high or uncontrolled for some reason at the gene transcription level. It is also possible that protein aggregation of TEV-C

**Fig. 4 | Amelioration of epileptic seizure by ST-Cal-Light. a** Schematic drawing of virus injection and fiber optic implantation. ST-Cal-Light viruses with TetO-EGFP reporter were injected into both hippocampal CA1 and CA3 areas bilaterally. **b** Representative images of tdTomato (transfection marker) and EGFP reporter gene expression. When seizure was induced by KA administration, little gene expression was created in the absence of light, demonstrating cell labeling was dependent on blue light. Scale bars, 50 μm. **c** A box-and-whisker plot of G/R ratio. Each circle indicates the cell (Seizure only: $0.258 \pm 0.005$ from 252 cells; light + seizure: $0.629 \pm 0.017$ from 252 cells, $P = 6.71 \times 10^{-72}$). The top and bottom of the box indicate the 25th and 75th percentile, respectively, the horizontal line across the box presents the median, and the whiskers mean the minimum and maximum values. Asterisks (***$P < 0.005$) indicate a two-tailed unpaired $t$-test. **d** For inhibition experiments, a TetO-NpHR reporter was used. The same ST-Cal-Light viruses were injected, as shown in (**a**). **e** A cartoon demonstrating experiment procedures. A seizure was induced by KA injection, and blue light was illuminated for labeling.

Two days later, KA was injected again for the second seizure induction and compared the severity of the seizure with or without 589 nm light. **f** Sample movement traces after KA injection with or without yellow light. Movement during the same period of time was plotted. Different colors represent animal movement speed. ezTrack analysis was used for tracing (from Denise Cai's lab)[44]. **g** Time-lapse changes of seizure score after KA administration. **h** Average seizure scores at various conditions. The random blue label condition was 5 s of blue light 3 times at the 1 min interval during the fear conditioning. **i** Hippocampal granule cells (GCs), mossy cells (MCs), CA1 and CA3 neurons were labeled by ST-Cal-Light, indicating increased neuronal activity in broad hippocampal areas. Scale bars, 50 μm. **j** Representative traces of normal and ictal-like activity before and after KA injection. **k** Time course of seizure activity progression after KA injection. The data of the time course are shown as mean values ± SEM. **l** Average changes in seizure score over time. **m, n** Comparison of seizure activity period between the first 10 min and the rest of 30 min. Source data and statistics are provided in the Source Data file.

and TEV-N is made in the subcellular region, which may cause excessive gene transcription.

In the case of transient behaviors, capturing involved neurons requires a high induction rate with low background signals. Our ST-Cal-Light is designed to cause the Cal-Light protein expression to condense in the cell soma; the chance to cause protein–protein interaction (two Cal-Light components) increases upon each blue light exposure. Therefore, a sufficient level of reporter gene expression can be obtained via a small number of light repeats. We tested context-dependent fear conditioning because it can be achieved by a few trials with very strong stimuli. We found that just three rounds of blue light exposure were sufficient to express NpHR in labeled neurons which, when activated, was sufficient to inhibit the fear memory. Presumably, $Ca^{2+}$ levels were robustly enhanced by electric shock, increasing the efficiency of gene expression.

Tagging neurons involved in social interaction is also a good candidate for modeling transient behaviors. The time for a single social interaction event is brief but repetitive. The frequency of interaction is also variable between animals. Thus, dissecting out only active cells and testing behavioral causality could be performed on individual animals, and the correlation between labeling efficacy and behavior could be compared. Because social interaction is not simply mediated by neurons in a single brain area, but by multiple circuits[32–36], viral injections in several brain areas will target involved neurons more comprehensively, and optogenetic reconstruction of full social behavior may be possible.

Generation of an ST-Cal-Light KI mouse can improve the degree of specificity by cell type when it is combined with other specific cre driver lines. The development of the ST-Cal-Light mice also reduces variability associated with virus injection. Tagging active neurons only from a specific cell type will give a better understanding of circuit functions. We confirmed that the conditional ST-Cal-Light mice limited gene expression only to specific cell types. Thus, the ST-Cal-Light system allows investigators to dissect neural circuits into individual cells and test direct behavioral causality in space and time.

## Methods
### Ethical statement
Research complies with all relevant ethical regulations. All experimental procedures and protocols were conducted with the approval of the Max Planck Florida Institute for Neuroscience (MPFI) Institutional Animal Care and Use Committee (IACUC), Johns Hopkins University IACUC, and National Institutes of Health (NIH) guidelines.

### Construction of plasmids
Entire DNA sequences of constructs used in this study are described in the Supplementary information. Concisely, to generate pAAV:: CMV-FLEX-TM-KA2-CaM-TEV-N-AsLOV2-TEVseq-tTA, pCMV::TM-KA2-CaM-NES-TEV-N-AsLOV2-TEVseq-tTA, and pCMV::TM-KV2.1-CaM-

NES-TEV-N-AsLOV2-TEVseq-tTA, we acquired DNA sequences for CaM, TEV-N, AsLOV2, and tTA from pAAV::TM-CaM-NES-TEV-N-AsLOV2-TEVseq-tTA (Addgene #92392) by the conventional PCR reactions. Amplified PCR products and digested pAAV or pCMV empty backbone were cloned in the overlap cloning technique using overlap cloner (ELPIS-BIOTECH, CAT #EBK-1012). Similarly, pAAV::M13-TEV-C-IRES-SP6-Cre was constructed by inserting amplified M13 and TEV-C, and synthesized IRES-SP6-Cre into the pAAV empty vector. To build pCMV::TM-KA2-CaM-NES-TEV-N-AsLOV2-TEVseq-tTA and pCMV::TM-Kv2.1-CaM-NES-TEV-N-AsLOV2-TEVseq-tTA, both KA2 and Kv2.1 were synthesized in Macrogen and sequentially digested by XhoI, and each fragment was inserted between TM and CaM domain of pCMV::TM-CaM-NES-TEV-N-AsLOV2-TEVseq-tTA. NpHR-EYFP which was amplified from Addgene #20949, and hChR2 (H134R) from Addgene #20297 using the conventional PCR. Amplified fragments were inserted into AAV backbone vector respectively to clone pAAV::TRE-NpHR-EYFP. All restriction enzymes and reagents for cloning were purchased from New England BioLabs and ELPIS-BIOTECH (Republic of Korea). Cloned plasmid vectors were carefully confirmed by DNA sequencing analysis (Macrogen).

### Primary neuron culture and DNA transfection
Primary dissociated neuron cultures were performed as described in previous literature[8]. To briefly describe the process, CD IGS rat hippocampus (embryonic day 18–19) were quickly dissected and digested in 0.25% trypsin–EDTA (Invitrogen) at 37 °C for 8–10 min. Trypsin-EDTA was then removed, and the hippocampal brain tissue was gently triturated ~10–15 times using a 100–1000 μL pipette tip. Twelve millimetre PDL-coated coverslips (Neuvitro) were placed on 24-well plates. Dissociated cells were counted and plated with a $10^5$ cells concentration in each well. The medium used to plate neurons consisted of a neurobasal medium (Invitrogen) with 1% FBS (Thermo Fisher Scientific), 1% Glutamax supplement (Gibco), and 2% B27 supplement (Gibco). Every 3–4 days, one-half of the media was replaced with freshly prepared medium lacking FBS. On DIV 9, cultures were transfected with DNA constructs using a Lipofection method (Thermo Fisher Scientific, Lipofectamine 3000 Reagent, #L3000008) for sparse DNA transfection. Diluted DNA constructs were mixed into the Lipofectamine reagent at a 1:1 ratio for each membrane specific receptor (pCMV-TM-*KA2 or Kv2.1*-CaM-NES-TEV-N-AsLOV2-TEVseq-tTA and M13-TEV-C-P2A-tdTomato). On DIV 14, all neuronal cultures were fixed and imaged using a confocal microscope (Zeiss LSM880).

### Myc tag staining in vitro
Myc (to visualize soma targeted receptor) and tdTomato (for M13-CaM transfection confirmation) staining for neuronal cultures was accomplished as follows: individual cultures were rinsed three times in PBS, pH 7.4; slices were blocked in 10% normal donkey serum (Jackson ImmunoResearch, 017-000-121) and 0.1% Triton-X for 30 min and

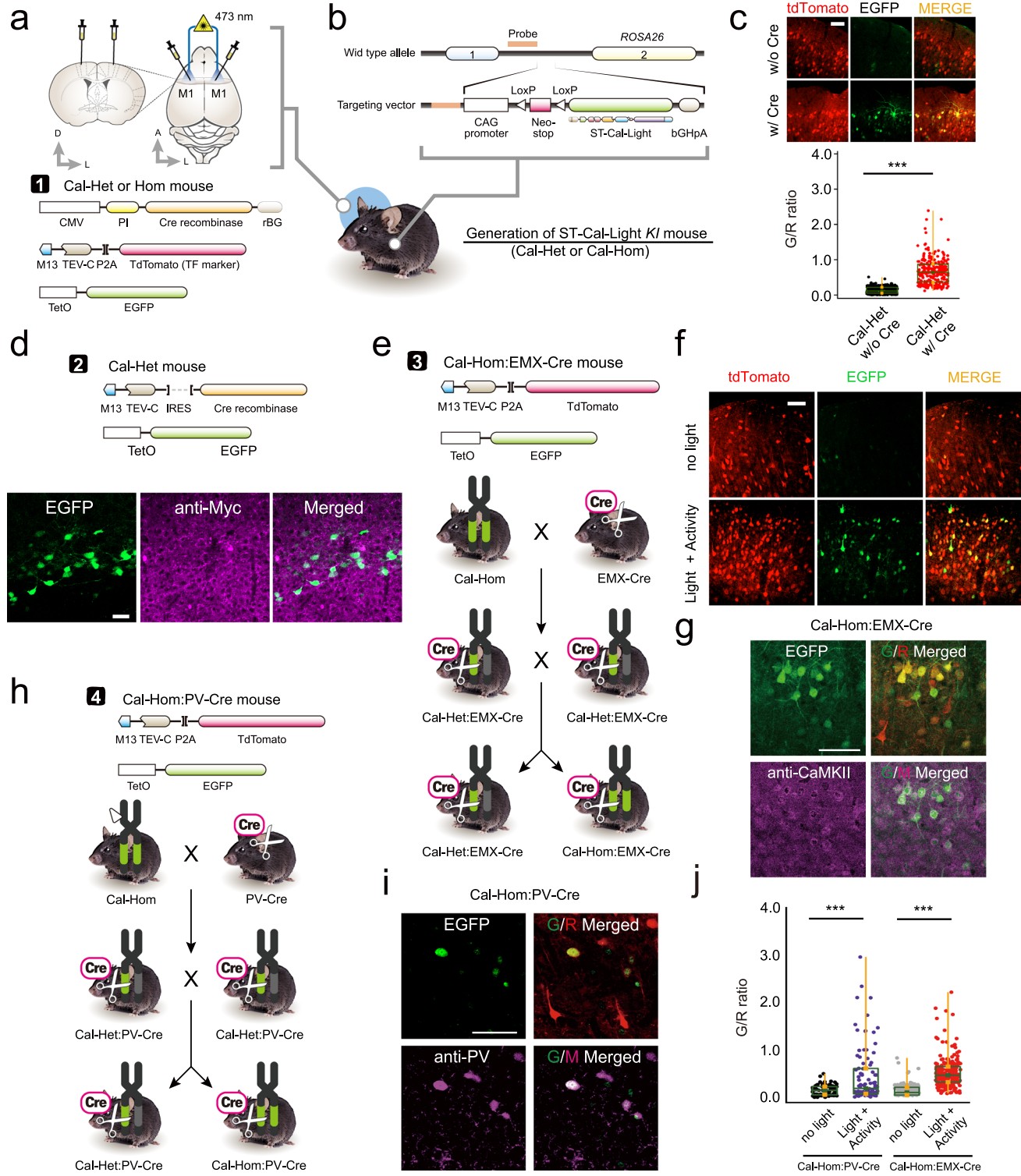

**Fig. 5 | Generation of conditional ST-Cal-Light knock-in mice. a** Schematic illustration of virus injection and fiber optic implantation. **b** Plasmid design for generating conditional ST-Cal-Light KI mouse. **c** Representative images of tdTomato (transfection marker) and EGFP (reporter) and G/R ratio distributions with or without introducing Cre recombinase (Cal-Het w/o Cre: 0.137 ± 0.005 from n = 292 independent cells; Cal-Het w/ Cre: 0.671 ± 0.022 from n = 258 independent cells). Scale bar, 100 μm. The border lines of the box indicate the 25th and 75th percentile, respectively, the horizontal line in the box shows the median and the whiskers mean the minimum and maximum values. Asterisks indicate ***P < 0.001. **d** Injected viruses (top) and images of EGFP and anti-myc staining. Myc epitope is expressed in a cre-dependent manner and localized at the somatic membranes. EGFP signals

indicate active neurons. Scale bars, 50 μm. **e** Cartoon of the breeding scheme. **f** A mixture of viruses was injected into the primary motor cortex of Cal-Hom:EMX-Cre mouse (top). Blue light-dependent gene expression in neocortical excitatory neurons under Emx1 promoter (bottom). Scale bars, 50 μm. **g** Excitatory neuron labeling was confirmed by CaMKII antibody staining. Scale bar, 50 μm. **h** Virus injection scheme in Cal-Hom:PV-Cre mice. **i** Schematic flow of generating either Cal-Het:PV-Cre or Cal-Hom: PV-Cre (left). Active PV-positive neurons were labeled and confirmed by PV antibody staining. Scale bars, 50 μm. **j** Cell-type specific, light- and activity-dependent gene expressions were confirmed. Source data are provided as a Source Data file.

inserted into a shaking incubator set to 23 °C at 120–130 RMP; incubated with a mixture of RFP antibody pre-absorbed (1:1000 in blocking reagent, Rockland antibodies & assays, 600-401-379) and Anti-Myc tag antibody (1:1000 in blocking reagent, Abcam, #ab32) for 90 min in room temperature shaking incubator; rinsed in PBS three times with 5-min incubation periods each time; incubated in Cy3-AffiniPure Donkey Anti-Rabbit IgG (H + L) (1:500, Jackson ImmunoResearch, 711-165-152) and Alexa Fluor 488 AffiniPure Donkey Anti-Mouse (1:500, Jackson ImmunoResearch, 715-545-150) for 30 minutes at room temperature shaking incubator; rinsed with PBS three times with 5-min incubation periods; mounted with DAPI Fluoromount-G (Southern BioTech, 0100-01). All cultures were imaged with a confocal microscope (Zeiss LSM880).

## Measuring the brightness of the fluorescence along the somatodendritic axis

The image analysis was performed through ImageJ software. After defining the boundaries of the soma from each neuron, a 20 μm diameter circle near the soma was made where no apparent fluorescence exists. The average fluorescence in this circle was used as background fluorescence. The boundary of the soma was defined by using the polygon tool, and the average fluorescence inside of it was measured and subtracted from the aforementioned background value. This value was defined as 'soma fluorescence.' To measure fluorescence intensities along the dendrites, a 1 μm² rectangle on the somatodendritic axis was made at every 10 μm from the soma and measured up to 100 μm. The distance between each rectangle and the soma did not correspond to the minimal linear distance from the soma (since dendrites were curved). To calculate the ratio of fluorescence intensity at each position, the fluorescence intensity at each rectangle was measured, and the background was subtracted. This value was divided by the soma fluorescence and plotted as a function of distances along the dendrites.

## Preparation of cortical organotypic slice culture and virus infection

Organotypic slice cultures were made from P2-P4 C57BL/6 mice (Charles River Laboratory). The general procedures for organotypic slice cultures were performed as previously described[8]. Coronal sections of the cortex (thickness, 400 μm) were made by a tissue chopper (The Mickle Laboratory Engineering Co. Ltd., UK). The age of culture is indicated by an equivalent postnatal (EP) day; a postnatal day at slice culture (P) + days in vitro (DIV). On EP 5-8, cultured slices were infected by adding a 5 μl of a solution containing 1 μl of concentrated virus (titer: ~$10^{13}$–$10^{14}$ GC/ml) and 4 μl of slice culture media (pre-warmed at 37 °C) to the top of the cortical layers of brain slice placed on porous (0.4 μm) membrane (Millicell-CM; Millipore) for covering the whole slice to maximize the infection rate. After infection, all the 6-well dishes that contain cultured slices were returned to the incubator (37 °C) with the aluminum foil covered to prevent unexpected light exposure. Experiments were performed at EP 20, two weeks after the viral infection. ST Cal-Light viral AAV vectors were cloned in the lab, and viruses were produced at ViGene Bioscience (Rockville, MD, USA).

## Activity- and light-sensitive neuronal tagging in organotypic slice cultures

Cultured slices were taken out from the incubator for induction experiment and superfused with autoclaved artificial cerebrospinal fluid (ACSF) solution containing the followings (in mM): 124 NaCl, 26 NaHCO₃, 3.2 KCl, 2.5 CaCl₂, 1.3 MgCl₂, 1.25 NaH₂PO₄, 10 glucose, saturated with 95% O₂ and 5% CO₂ gas. The concentric bipolar electrode (12.5 μm inner pole diameter, 125 μm outer pole diameter; FHC Inc.) was positioned at layer 2/3 cortical area. Stimulation pulses (100 μs in duration, stimulus intensity, 10–15 V) were generated by a digital stimulator (Master 8, AMPI, Israel) and fed into the stimulation

electrode via an isolation unit (DS2A, Digitimer Ltd.). A blue laser (MBL-F-473 nm–200 mW, CNI, China) coupled to an FC/APC fiber (400 μm, 1-m long, CNI) was positioned two centimeters above the surface of the cortical slice. The total power from the tip of the fiber was 5–10 mW. The induction protocol used in this study elicited about 75 spikes. This number was much lower than one used in a previous study (~900 spikes)[8,31]. The blue light was illuminated for 15 min (10 s ON/50 s OFF) with or without electrical stimulation (Light + Activity vs. Light only). One train (5 pulses at 10 Hz) of electrical stimulation was delivered per min with 10 s-long blue light. Sample whole-cell patch–clamp recordings were made to measure the number of spikes that elicited by the weaker induction protocol through a MultiClamp 700B amplifier controlled by Clampex 10.2 via Digidata 1440A data acquisition system (Molecular Devices). The pipette solution was made as follows (in mM): 125 K-gluconate, 5 KCl, 10 Na₂-phosphocreatine, 4 Mg-ATP, 0.4 Na-GTP, 10 HEPES, 1 EGTA, 3 Na-ascorbate (pH = 7.25 with KOH, 295 mOsm). For G/R analysis, we included all neurons at each experimental condition, and always green and red fluorescence levels were analyzed in a consistent manner without adjusting background intensity. Red signals were similar across all conditions.

## Animals

Experimental subjects were used from 6 to 12 weeks old C57BL6J mice (Jackson Laboratory, Bar Harbor, ME, USA, Stock No: 000664). Similar numbers and ages of both male and female mice were randomly chosen for experiments. All mice for behavior tests were individually housed in a 12-h dark–light reverse cycle, and experiments were performed during the dark cycle period. Mice born into our colony on a C57BL6J background were maintained in conventional housing with free access to food ad libitum when not being tested. Cre mouse lines used in this study were purchased from Jackson Laboratory: EMX1-Cre (Stock No: 005628), and PV-Cre (Stock No: 017320).

## Stereotaxic surgeries

Viral injections and fiber implantations were performed as described in a previous study[8] with the following specifications. For targeting primary motor cortex (M1) area, virus-containing solution (pAAV-pCMV-Myc-TM-KA2-CaM-NES-TEV-N-AsLOV2-TEVseq-tTA: pAAV-hSYN-M13-TEV-C-P2A-tdTomato: pAAV-TRE-EGFP = 2: 2: 5 ratio) was injected for the labeling of the active population of neurons at AP, + 0.25 mm; ML, +1.5 mm from bregma, and DV − 0.2 mm from the brain surface or pre-mixed viral solution (pAAV-CMV-Myc-TM-KA2-CaM-NES-TEV-N-AsLOV2-TEVseq-tTA: pAAV-hSYN-M13-TEV-C-P2A-tdTomato: pAAV-TRE-eNpHR-EYFP, 2: 2: 5) was injected to the same target site for the manipulation of the labeled subset of neurons. After viral injection, the optic fibers (200 μm core; NA 0.37; Cat# BFL37-2000, Thorlabs) were implanted perpendicularly into the targeted brain region. The tip of the fiber was positioned 100–150 μm above the target viral injection site in the M1 area. To label and manipulate task-relevant neuronal populations in hippocampal CA1 neurons during the contextual fear conditioning task, the same 500 nl mixture of AAV viral solution was injected into the dorsal hippocampal CA1 region at the following coordinate (AP, − 2.0 mm; ML, ± 1.3 mm; DV, − 2.05 mm from bregma) and the optic fibers were implanted 300 μm above injection site to deliver the light down to the target brain region. For the social cognition experiment, we bilaterally injected either of pAAV-CMV-Myc-TM-KA2-CaM-NES-TEV-N-AsLOV2-TEVseq-tTA: pAAV-hSYN-M13-TEV-C-P2A-tdTomato: pAAV-TRE-EGFP (for labeling) or pAAV-CMV-Myc-TM-KA2-CaM-NES-TEV-N-AsLOV2-TEVseq-tTA: pAAV-hSYN-M13-TEV-C-P2A-tdTomato: pAAV-TRE-eNpHR-EYFP (for manipulation) into the mPFC area (AP, + 2.0 mm; ML, ± 0.5 mm; DV, − 1.6 mm from bregma) and the optic fibers were implanted into the same hole of virus injection at 5–10° angle. The tip of the fiber was placed 500–700 μm above the sites of the virus injection using a cannula folder (Thorlabs). For the validation of efficiency in distance-dependent labeling, AAV mixture of the ST-Cal-Light was

injected into the mPFC area (AP, + 2.0 mm: ML, ± 0.5 mm; DV, −1.6 mm from bregma), and the fiber tip was implanted at 400 μm and 800 μm above the virus injection site. For in vivo seizure experiments, multiple injections per hemisphere were targeted bilaterally to maximize the number of neurons responsible for the seizure activity (CA1: AP, −1.8 mm; ML, ±1.5 mm from bregma, and DV −1.7 mm; DG: AP, −1.9 mm; ML, ±1.1 mm; DV −1.8 mm from bregma). To label the broader areas and the numbers of neurons during a seizure, multiple optic fibers were implanted over the hippocampus at (relative to Bregma): AP, −1.8 mm; ML, ±1.3 mm; DV −1.2 mm. To relieve post-surgical pain, an analgesic (buprenorphine, 0.05 mg kg$^{-1}$ of body weight) was injected subcutaneously, and mice were returned back to their home cage for recovery.

### Optical labeling of the task-related neurons
Three weeks after viral injection, blue light (473 nm) was delivered to the regions of interest (M1, mPFC, or hippocampus) during specific behavior. Blue laser (MBL-FN-473, Changchun New Industries Optoelectronics Technology, Jilin, China) was controlled by MED-PC IV software, which connected with an acquisition board (Med-Associates, St. Albans, VT).

### Skilled motor learning
Water restriction was started 3–4 days prior to the lever-press training, and water was restricted throughout the training periods, so the mice receive the rest of the water after subtraction of the amount of water given during training to make a total of 1 ml of water per day. The training was performed in a standard mouse operant chamber (Med-Associates, St. Albans, VT) placed in a sound-attenuating cubicle (ENV-022MD, 22 cm × 15 cm × 16 cm). In the continuous reinforcement (CRF) session, a mouse received a water reward provided from a retractable sipper tube extended into the chamber after lever press, and they learned that lever pressing was associated with a water reward. During the CRF 5Rs and CRF 10Rs, mice received five rewards and ten rewards on each training day, respectively. When mice finished getting fifteen rewards (CRF 15Rs) on training day 5, they were moved to the fixed ratio (FR) schedule, which consists of FR-2 (2 presses to get a reward), FR-5, FR-8, FR-10, and FR-12 (lever pressing 12 times to get the reward once) for 5 consecutive days. These FR sessions lasted 45 min or until mice received 20 rewards. Labeling was performed by two separate protocols. For full labeling, 473 nm blue light (5 s ON/25 s OFF cycle) was illuminated from CRF 15Rs to FR-12. For mild labeling, blue light with the same cycle was delivered from FR-8 to FR-12. The light-only control experiments received blue light for 600 s (5 s ON/25 s OFF, 120 times) under anesthetization.

### Contextual fear conditioning test
The contextual fear conditioning task was composed of three experimental phases (habituation, fear acquisition, and fear retrieval) and conducted on two consecutive days in large sound-proof isolation chambers (Med-Associates, St. Albans, VT). On day 1, mice underwent the habituation phase for 10 min in a neutral context. The mice were then subject to the fear acquisition phase for 5 min, in which the neutral context was paired with an electric foot shock (2 s, 0.7 mA) delivered at 120, 180, and 240 s, respectively. To label and subsequently manipulate a population of neurons activated during the context-shock association, blue light (5 s) was delivered while mice receive foot-shock 3 times with a 1 min interval. Two days after fear acquisition, for the retrieval of contextual fear memory, the mice were re-exposed to the conditioning context, and freezing responses were measured for 5 min.

### Sociality test
The sociality test was performed in an open field chamber (42 × 42 × 42 cm, W × D × H) as shown previously[37,38]. The test was

composed of labeling sessions for two consecutive days, and a probe test was performed 48 h after the second labeling day. During a 5-min habituation session, test mice freely explored in the open-field chamber where two empty plastic cylinders (10 × 10 × 18 cm, W × D × H, left and right upper corner) are located at the corner. After the habituation, a novel mouse (the same sex and age-matched C57B6/J mouse) was placed in one of the chambers (social chamber), and a mouse-shaped toy was placed in the other chamber (object chamber). For labeling, a blue laser (5-s duration) was delivered when the test mice enter a 3 cm area around the social chamber. The laser was terminated when the test mice stayed in the area for more than 5 s. The sociality test was performed 48 h after the last day of labeling sessions. After the 5-min habituation, test mice explored the open-field chamber with both social and object chambers for 10 min. A 589 nm yellow light was illuminated to the test mice during the next 10 min, and sociality was compared. The behavior was recorded through a megapixel 720p USB camera (Shenzhen Ailipu Technology, China). The nose and both sides of the ear were tracked by DeepLabCut[39]. Staying time around each chamber was quantified as the total time when the nose or the ears were within 4 cm from the boundary of the chamber. The social preference index was measured as follows; Social preference index = Time around the social chamber/(Time around the social chamber + time around the object chamber). The blue and yellow lasers were controlled by Pulse-Pal (Sanworks, NY, USA) and a custom-made python code.

### Seizure activity
Mice were injected i.p. with KA (20 mg/kg) to induce an acute seizure. This dose was used to induce generalized seizures but was sublethal[40]. After 10 min, blue light was delivered (3 s ON/2 s OFF; for 30 min) into the hippocampal region through the bilateral optic fibers to label the neuronal population, which is engaged in seizure activity. Seizure severity was classified and scored according to a modified Racine score:[41–43] 0, normal; 1, immobilization, sniffing; 2, head nodding, facial and forelimb clonus (short myoclonic jerk); 3, continuous myoclonic jerk, tail rigidity; 4, generalized limbic seizures with kangaroo posture or violent convulsion; 5, continuous generalized seizures (clonic–tonic convulsions); 6, death. The seizure-only control group was not received blue light during the seizure, but virus injection and fiber optic implantation were made in the same way as for the test group.

### Behavioral manipulation by optical inhibition or reactivation of task-relevant neurons
Mice were allowed to recover from the viral injection (pAAV-CMV-Myc-TM-KA2-CaM-NES-TEV-N-AsLOV2-TEVseq-tTA: pAAV-hSYN-M13-TEV-C-P2A-tdTomato: pAAV-TRE-eNpHR-EYFP, 1:1:2 ratio) and optic fiber implantation surgery. For optical inhibition of skilled motor learning, a probe test (589 nm, 2 s ON/1 s OFF; for 45 min) was conducted on the third day after the last training session. The power of the laser was measured at the end of the tip of an optic fiber and adjusted to be 15–20 mW as described previously[8]. For optical inhibition of contextual fear memory, yellow light illumination (3 s ON/2 s OFF) began when mice were re-exposed to the same context 48 h after fear acquisition. Contextual fear retrieval was measured by recording freezing responses to the conditioning context for 5 min. Freezing (%) was scored every 10 s by two researchers blinded to the group assignments, with their scores being cross-checked with correlation analysis. For optical reactivation of fear memory, a mixture of AAVs expressing ST-KA2, M13-TEV-C, and TetO-ChrimsonR-EGFP, respectively, was injected into the dorsal hippocampus (CA1 area) bilaterally. Two days after contextual fear conditioning, mice were placed in a novel context (context B) distinct from the conditioning context (context A). The floor and walls of the novel context were constructed with white plastic plates, with them being wiped with 0.5% acetic acid prior to the introduction of each individual mouse. Freezing behavior

in context B was recorded for 6 min in duration, consisting of the first 3 min light-OFF epoch and the last 3 min light-ON epoch. During the light-ON epoch, ChrimsonR was stimulated at 20 Hz (5-ms pulse width) using a 589 nm laser.

For optical inhibition of social cognition behavior, the labeled social cognition-related neuronal population was inhibited by the delivery of yellow light (3 sec ON/2 sec OFF) through the bilateral optic fiber on the second day after the blue light labeling. The number of social interactions and time around chambers were analyzed as behavioral readouts. For optical inhibition of seizure-responsive neurons, mice received the second KA injection to re-induce the seizure. 589 nm yellow light was illuminated into the same hippocampal area to inhibit neural activity (by activation of NpHR). The mean seizure score and percent of the time in tonic seizure were compared between the first- and the second-day KA-injected mice, and mice groups in the presence/absence of yellow light. To label hippocampal neurons that are independent of seizure, blue light was illuminated when mice were receiving footshock in the chamber (the same contextual fear conditioning box). Bilateral 589 nm light was delivered into the hippocampus of fear memory-labeled mice (3 s ON/2 s OFF) during a seizure induced by the KA injection. The seizure scores were analyzed.

### Electroencephalogram (EEG) recording and seizure analysis

EEG signal was recorded using a wireless EEG device, Neurologger 2 A, to measure the seizure activity induced by KA while performing optogenetic stimulation. EEG electrodes were implanted along with optic fiber. After isoflurane anesthesia, the skin was removed, and a small burr hole was made with a dental drill over the skull of the hippocampus (1 mm lateral and 0.5 rostral to the bregma). An adapter that is wired with microscrew electrodes was mounted. One electrode was implanted over the hippocampus for recording and another one over the skull of the cerebellum for reference and ground. After 5–6 days of recovery, Neurologger 2 A device was mounted onto the adapter, and EEG signals were recorded before and after KA administration (1000 Hz, 4 times oversampling). For analysis, we used a custom written python script to visualize signals and frequencies to mark the seizure events. With 2.5 seconds binning, the time bin was marked as seizure when there are abnormally high amplitudes (~2 times higher) or apparent changes in the frequency range from the normal range. The duration of the seizure was calculated every 5 min for 40 min after KA administration and presented as a time course graph.

### Tissue fixation and acquisition of confocal images

The general procedures for tissue fixation were performed as shown in a previous study[8]. Briefly, animals were deeply anesthetized by a mixture of ketamine and xylazine and then perfused transcardially, first with PBS (pH 7.4) and then with 4 % paraformaldehyde (PFA) dissolved in PBS. The brains were removed and postfixed in 4% PFA overnight at 4 °C. The brains were embedded into 10 % melted gelatin solution for 50 min at 50 °C, and then the gelatin solution was refreshed. The gelatin solution with the embedded brains was kept at 4 °C for ~30 min for solidification of the gel. The gel was trimmed to a small cube around the brain, and the cube was kept in a 4% PFA solution overnight. A coronal section (thickness, 100 μm) was made using a vibratome (Leica VT1200) for confocal imaging. Imaging was performed using an upright confocal laser-scanning microscope (LSM880, Zeiss, Germany) with a 20×/0.8 M27 objective lens. The green-to-red ratio (G/R) value of individual cells was analyzed using ImageJ (NIH).

### Immunohistochemistry

After labeling by ST Cal-Light, the mice were immediately perfused with 4% paraformaldehyde in phosphate buffer solution (PBS). The brain was removed and sliced with 100 μm thickness. Slices were washed three times in PBS and blocked with 10% Normal Goat Serum in PBS with 0.1% Triton-X (Sigma) for 1 h at room temperature (RT). Thereafter, the primary antibody was applied to the slices at 4 °C for 1–2 days (1:500 for PV and CaMKII). After washing the slices three times, the slices were incubated with secondary antibodies with Goat anti-Mouse IgG Alexa flour 633 (Thermo Fisher Scientific, #A21052) or Goat anti-Rat IgG Alexa flour 647 (Thermo Fisher Scientific, #A21247) (1:300 for 2 h at RT. To prevent the bleaching of fluorescent signals, VECTASHIELD Hard set with DAPI (Vector Laboratories, Inc. Burlingame, USA) was applied to the slices on a slide glass. The images were captured using confocal microscopy (Zeiss 800, Zeiss).

### Vector design for conditional overexpression of ST-KA2 in ROSA26 Locus

Plasmid design and ST-Cal-Light knock-in mice were generated by Ingeneous targeting laboratory (Ronkonkoma, NY). The vector was designed to have the expression of the TM-KA2-CaM-NES-TEV-N-AsLOV-TEV-tTA knock-in sequence under the pCAG promoter and also be controlled by a floxed stop cassette. The TM-KA2-CaM-NES-TEV-N-AsLOV-TEV-tTA sequence was cloned into the MluI site of the ROSA26-pCAG stop backbone vector using the conventional cloning method. The stop cassette consists of a floxed PGK/gb2neoPGKpolyA2XSV40pA sequence. The knock-in sequence is followed by BGHpA sequence. The targeting vector contains a short homology arm (SA) with a 1.08 kb ROSA26 genomic sequence upstream of the pCAG promoter and a 4.34 kb long homology arm (LA) downstream of BGHpA sequence. The targeting vector was confirmed by restriction enzyme analysis and sequencing after each modification. The boundaries of the pCAG-stop cassette and BGHpA-ROSA26 genomic sequences were confirmed by sequencing with primers ROSASQ1 and ROSASQ2.

In order to test the efficacy for labeling of an active subset of neurons using newly generated Cal-KI mice in vivo (Fig. 5), pre-mixed viral solution (pAAV-M13-TEV-C-IRES-SP6-Cre: pAAV-TRE-EGFP, 1:4) was injected to the M1 (Coordinates: AP, + 0.25 mm; ML, + 1.5 mm from bregma, and DV − 0.2 mm from the brain surface; see Fig. 5d). The optic fiber implantation coordinates for the labeling of M1 neurons in ST-Cal-Light KI mice is described above (see stereotaxic surgeries section).

### Generation and verification of double transgenic mice

Male and female ST-Cal-Light KIs and WT littermates were maintained on a C57BL/6 background. For identification of active PV interneurons, KI heterozygous (Cal-Het) mice were bred with Cal-Het::PV-Cre mice to generate Cal-Het (or Hom) and WT littermates that were PV-Cre hemizygous (see Fig. 5i). PV-Cre mice were derived from a mouse knock-in of Cre recombinase directed by the PV promotor/enhancer (Pvalbtm1(cre)Arbr, The Jackson Laboratory; Stock No: 017320). The same procedure for the generation of Cal-Het::EMX-Cre was used with the following verification. For targeting expression into either M1 pyramidal or PV neurons, 700 nl viral stock solution (pAAV-hSYN-M13-TEV-C-P2A-tdTomato: AAV1-CMV-PI-Cre-rBG: pAAV-TRE-EGFP = 1:2:4 ratio) was injected to the same target site for the manipulation of the labeled subset of neurons in Cal-Het::EMX-Cre or Cal-Het::PV-Cre mice, respectively (see above fiber implant coordination).

### Pharmacology

Bicuculline and TTX were purchased from Tocris Bioscience (Minneapolis, MN, USA). They were prepared as a stock solution in high concentration (1000× or greater), and the final concentration was met via dilution.

### Inclusion and ethics

Local researchers were included throughout the entire research process and followed the Global Code of Conduct for Research in Resource-Poor Settings.

## Statistics and reproducibility

The statistical significance of culture neuron data was calculated by one-way ANOVA with post hoc Bonferroni test using SPSS 21 (IBM) software. Organotypic slice culture and in vivo data were analyzed using IgorPro (version 6.10 A; WaveMetrics, Lake Oswego, OR, USA) and JMP Pro (version 16.0, SAS Campus Drive Cary, NC, USA). Statistical data were presented as means ± standard error of the mean (s.e.m., denoted as error bars), and $n$ indicates the number of cells or animals studied. For assessing the normality of data distribution, we performed the Kolmogorov–Smirnov test. For comparison between two independent samples, we applied a two-tailed Student's $t$-test (parametric) or Mann–Whitney $U$ test (non-parametric) after testing normality. For comparison between two conditions from the same samples, paired $t$-test was applied after confirming the normal distribution of data. For comparison of data from multiple samples that show normal distribution, ANOVA was applied. In particular, when the data was obtained in several conditions multiple times, two-way RM ANOVA was performed. Bonferroni correction was used for the correction of post hoc multiple comparisons. Data presented in Fig. 1b, d-f are confirmed in five independent cultures. Experiments shown in Figs. 2e, 3c, 3j, 4b, 4i, 5d, 5f, 5g, and Supplementary Fig. 4c, 4f are repeated and confirmed at least 5 mice. The difference was considered as significant when $P < 0.05$. n.s.: no statistical significance; $*P < 0.05$, $**P < 0.01$, $***P < 0.005$. No statistical methods were used to predetermine sample sizes. Mice that did not learn level-pressing behaviors during the first phase of training (days 1–4) were excluded from continuous training. All cell counting and behavioral experiments were performed by an experimenter blind to the experimental condition only, except for the ST-KA2-Cal-Light KI mice experiments. All subjects were randomly allocated to different experimental conditions.

## Reporting summary

Further information on research design is available in the Nature Portfolio Reporting Summary linked to this article.

## Data availability

All plasmids used in this study were deposited and available in Addgene (ID: 171615–171622). The conditional ST-Cal-Light KI mice are available from the corresponding author upon request. Delivery depends on the breeding condition and administrative processes, which are expected to take 6–8 months. Source data are provided in this paper.

## Code availability

Codes used in this paper are available online: https://github.com/knagahama1001/Hyungbae-Kwon-lab.

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

## Acknowledgements
We thank members of the Kwon laboratory for helpful discussion and Holly Robinson for manuscript proofreading. This work was supported by the National Institute of Health Grants DP1MH119428 (to H.-B.K.), NS036715 (to R.L.H.), R01MH053608 (to P.F.W.), MH-092443 (to A.S.), MH-094268 (to A.S.), MH-105660 (to A.S.), and MH-107730 (to A.S.); foundation grants from Stanley (to A.S.), and RUSK/S-R (to A.S.). This work was also supported by National Research Foundation of Korea (NRF) grants (NRF-2018M3C7A1024597) (to D.L.); the Korea Government Ministry of Science and Information and Communications Technology (ICT) (to D.L.); the DGIST R&D Program of the Ministry of Science and Information and Communications Technology (ICT) (22-PCOE-01to J.H.H.), and JSPS Overseas Research Fellowships (60-236) (to K.N.)

## Author contributions
H-B.K., J.H.H., and D.L. conceived and designed the study. J.H.H. performed experiments and data analysis related to lever pressing training, seizure induction, and conditional KI mouse verification. J.H.H. did data analysis from slice culture experiments. K.N. performed social interaction experiments, lever pressing training, seizure experiments, EEG recording, mouse breeding, and related data analysis. H.N. and A.S. performed experiments and data analysis related to context-dependent fear conditioning. N.M. performed dissociated cell culture experiments. S.-E.R. and P.W. performed EEG recordings. P.H. helped with surgeries and behavior experiments related to lever pressing. S.K. performed experiments using organotypic slice culture. M.C., S.L., and D.L. performed DNA plasmid construction and cloning. B.L. and R.H. performed the initial verification of myc antibody staining. A.M. helped slice culture data analysis. C.K. performed behavioral analysis using the ezTrack and DeepLabCut methods. H.-B.K. wrote the first draft and all authors edited the paper.

## Competing interests
The authors declare no competing interests.
