## [Peer Review File · Nature Communications]

Reviewers' Comments:

Reviewer #1:

Remarks to the Author:

Review signed by: Jesper Sjöström

Building on the elegant gene expression system (Cal-Light) that was previously reported by the same group (Lee and Kwon, Nat Biotech. 2017), Hyun and Kwon further refined the system by engineering the sensor to be soma-targeted. Briefly, this system serves as a coincidence detector of high calcium level and light activation. At high calcium, the membrane tethered sensor recruits a viral protease, which cleaves a viral linker that is only exposed by blue light activation. The cleavage process releases a transcription activator that promotes specific transgene expression in the nucleus. Since the goal of detecting high calcium lies primarily at sensing only active neurons that fire action potentials, the authors reasoned that targeting the sensor to soma would avoid off-target readout by reducing action potential-independent calcium sensing. This is particularly important in the in-vivo context of behaviours or memory-related circuits, where only the active cells within the desired time window can be precisely targeted and correlated with the task.

In cultured hippocampal neurons and cortical slices, Hyun et al. first verified that the new ST sensors are correctly enriched in the soma and enhance transgene expression up to two-fold compared to the original version. After confirming that ST sensors can be correctly induced by neuronal activity and light in vivo, the authors used the system to target halorhodopsin expression in neurons activated during four different behavioural paradigms. This showed expected behaviours to be inhibited when halorhodopsins were activated to depress neurons labelled by ST-Cal-Light, demonstrating the system effectiveness and utility in vivo. To partially circumvent the need to introduce multiple components of the ST-Cal Light system, the author generated a transgenic mouse line that expresses the sensor in a Cre-recombinase dependent manner. The mouse line was then verified for the ST-Cal Light system after crossing with two commonly used Cre lines.

The design of the new sensor with the soma-targeted motif addition is a potentially very important modification. The proof-of-principle behavioural experiments are well designed and performed. The results are convincing, with good controls for most of the experiments. Figures are of high-quality, although at times a bit compact, and the text is succinctly written, although more details in the Methods section would help and more explanatory text would help to clarify. The ST-Cal-Light mouse line will likely be a valuable contribution to the field. One caveat is possibly the leakiness of expression, the authors could be more upfront about this. After addressing these comments, we believe the present manuscript should be highly suitable for Nature Communications.

MAJOR POINTS:

1) Fig. 1h and 1k:

- a. Correlation coefficient and p values should be presented to support the correlation mentioned in the main text.
- b. How were the cutoffs (dotted lines) defined in the graph for B, G, Y and R quadrants? The y-axis cutoff actually changed from ~70 a.u. in Fig.1h to 100 a.u. Fig.1k.
- c. Presumably raw signal values are used for these plots, were the imaging conditions consistent across groups and experiment days? In our experience, virus injection does not always result in the same expression level between experiments. One quick way to avoid spurious patterns is to colour-code the datapoints by experiment days and ensure they don't form clusters by dates.

2) There appears to be a considerable proportion of cells with transgene expression but with no TdTomato reporter expression, in particularly in the Light + Activity condition and in vivo. This is suggestive of the M13-TEV-C-P2A-Tdtomato was not successfully transduced into these cells, and that the transgenes were expressed non-specifically in the absence of the TEV-C protease – a key component for Cal-Light to work correctly. For example, Fig. 2e, 3c, 4b, 4i, 5f, 5g and 5i. A couple of concerns arise here:

- a. How were the G/R ratios treated in these instances? (Also relevant to Fig.1g and 1j) The virtual

absence of R can result in the artefact of apparent large increases in the G/R ratio. Was a R signal threshold applied to select certain population of cells for quantification? How was this threshold determined? Without a good procedure, the G/R ratio can be skewed by using cells with (background) R signal coming largely from the neurites of other Tdtomato expressing cell.

b. Can the authors speculate why this is happening? It is highly advisable to check the detailed subcellular localization of the three components of ST-Cal-Light. One possibility is that some of the transmembrane sensors are mistargeted to the nuclear membrane instead of the cell membrane, allowing the TetO sequence in the nucleus to interact with tTA that is intact in the sensor, and in the absence of the TEV-C protease. While we appreciate an immediate remedy to the problem may not be possible, the caveats and speculations should be frankly noted in the discussion to help other users.

c. To estimate the leakiness of the ST-Cal-Light, quantification should be performed and presented in the manuscript to account for the percentage of cells with G signal but almost no R signal.

d. G/R ratio quantifications are missing for the the Cal-Hom: EMX-Cre and Cal-Hom: PV-Cre experiments.

3) To improve the impact of the ST-Cal-Light system, it would be important to compare how the new system with soma-targeting outperforms the old Cal-Light.

a. A comparison between Fig. 2 and their previous study (Lee and Kwon, Nat Biotech. 2017) seem to show very similar readouts with minimal improvement – G/R ratio, Lever press per min, Cumulative lever press.

b. It is possible that the above longer term labelling paradigm does not fully exploit the faster labelling capability of ST-Cal-Light. The manuscript would benefit from comparing old vs. new system in the fear conditioning paradigm (short labelling time), in which the total blue light exposure may reveal the improved efficiency.

4) The manuscript is not always as clear as it could be. Although Fig 1a clarifies, it wasn't immediately clear from the text how Cal-Light works and how light is involved, please describe on page 5, top. In general, the manuscript can be strengthened by including more details on procedures and analyses in the Methods section. It was not immediately easy to understand what was done with each experiment or figure unless one carefully reads the main text, figure captions, and methods section together. The light is an important component for the temporal resolution so it would be helpful to elaborate on that in the abstract. In general, figure captions should state what the figure means or what the reader should conclude, not what is in the figure, e.g. Fig. 2 "Confocal images showing EGFP expression at each condition" could be rephrased, for clarity's sake. Same thing with "Average of lever press number per minute under condition" and "Cumulative lever press number plotted against time," because this is not very informative, as the reader can already see this from the figure.

MINOR POINTS:

1) Fig.1i is not referred to in the main text

2) Fig. 2l- Significance asterisks missing between 589 OFF and 589 ON Full label conditions?

3) Fig. 2j- Meanings of the colour bars underneath the FR-12 label are not clear, please define in figure caption

4) Fig. 4h- The tiny orange strand in black colour scheme for the "2nd KA inj" condition is not apparent. The orange colour implies the 589 laser is ON when it is not.

5) Fig. 4i- Top row labels are colour-coded but not at elsewhere in the manuscript, the authors may want to standardize them to improve consistency.

6. The figure captions could be improved with more detail. For example, in figure 1j and figure 2g, it would be useful for the readers if data values were included (unless if it goes against journal guidelines). A good example would be the caption of figure 1g. Another example is the caption for figure 3i, could it be clarified whether each data point corresponds to one mouse?

7. It was mentioned that mice moved to FR sessions of the lever pressing motor learning task after obtaining 15 successful rewards in the CRF sessions. It is unclear to me in Figure 2b how this happens in Day 5 because the illustration tells me that it would be after 25 rewards, so perhaps some clarification here would be helpful.
8. Figure 2d was not cited in the main text.
9. Figure 3k is cited before 3j in the main text.
10. Introduction paragraph 2: what are some examples of the high cognitive behaviors? Consider rephrasing as higher-order cognitive behaviors.
11. Introduction paragraph 3: Not sure how non-specific signals would accumulate over time while waiting for viral expression. There would be no light, right? Is it from construct remaining Ca-bound?
12. Results: Since you spelled out kainate receptor 2 (KA2), it might be useful to also indicate Kv2.1 as a potassium channel.
13. Results end of paragraph 1: what is the "other partner"?
14. Perhaps the brightness/contrast in Fig 1B bottom panels can be adjusted? Right now, it is hard to tell the OG-Cal expression in dendrites
15. Figure 1h - Since ST-Kv2.1 and ST-KA2 plots are already indicated in the figure, they could perhaps be color-coded in the same way as Figure 1g - according to experimental condition.
16. Figure 2b - unclear what 5Rs, 10Rs, 15Rs mean
17. While figure 2c is nice, I don't feel like it accurately represents the 5 sec ON/25 sec OFF cycle). Consider revising or including it in fig 2b?
18. How is NpHR expressed? This method is very interesting so I think you should elaborate more in the text. Also, it is surprising that NpHR only needed 2 days to be sufficiently expressed to show a behavioural outcome.
19. Figure 2j, k: What about off-target expression of NpHR? Is there a no label condition (no blue light)?
20. Page 8 "reflecting the erasure of memory linking lever pressing to rewards" - I don't think this is true. Since you are inhibiting activity in M1, it is possible that the motor function was inhibited but the memory is not erased. I also don't think the memory was erased, because the day after 589 ON, the behaviour was restored. So you are inhibiting the behaviour (most likely scenario) or preventing the recall of the trained behaviour. Please clarify.
21. Page 9, were the foot shocks paired with an unconditioned stimulus? What was the procedure during the retrieval phase?
22. Figure 4g - consider changing the light grey line to a darker color
23. Figure 5i - it looks like cells are not co-expressing EGFP and PV? Is there further cell-counting data?
24. Page 6, "To verify whether similar results are also obtained in brain slices", please specify that these are not acute slices.
25. Page 7 and p9, "shined" should be "shone"
26. Page 7, "To compare with a weak labeling...", please clarify the rationale for this comparison.

27. Page 7, "... to match the same conditions in which ...", please clarify which conditions.
28. Page 8, "NpHR" in the text but "eNpHR" in the figure, please be consistent.
29. p8, typo, "the number of lever pressES"
30. p8, "Once mice finished FR12 training, 589 nm..." Does 589 nm light not induce Cal-Light? Please show.
31. p8, "not due to the tissue damage", please remove "the"
32. p8, "reporter was used as a negative control," (comma missing)
33. p8, "ST-KA2, M13-TEV-C, and TetO-NpHR-EYFP", please explain in text what each one does, for clarity.
34. p10, "It is generally believed that epileptic seizure is caused by unobstructed", unclear what "unobstructed" means here, please rephrase.
35. p10, "gene expression was only present in the group ...", Can we get a cell count too, pls?
36. p10, it might be useful to cite and very briefly discuss other optogenetics approaches in epilepsy research, to contextualize the pros and cons of the present approach.
37. p11, please remove "of the chromosome"
38. p11, "A LoxP flanked neo cassette was placed in the upstream of target gene" should be "A LoxP flanked neo cassette was placed upstream of the target gene"
39. p11, typo "blue lights"
40. p12, "into the layer2/3", please remove "the"
41. p12, several typos, see uppercase here: "tdTomato signal WAS driven by CAG promotEr, so its expression was not restricted TO specific cell types, but EGFP expression was induced in excitatory neurons because Cre expression WAS limited TO neocortical excitatory neurons under THE Emx1 promoter"
42. p12, poor grammar, also unclear: "Thus, conditional ST-Cal-Light mice specify active neurons from a distinct cell type."
43. p13, generally unclear: "because somatic action potentials are typically final output signals of neurons, their inhibition has a more prominent effect on behaviors" Should 'typically' be 'always'? etc
44. p13, "while less active neurons DO not"
45. p13, "Labeling efficiency ... ", this sentence is unclear. Also, it should be in past tense, because this is a reported finding, right?
46. p13, these three sentences are generally unclear and hard to follow: "In case of transient behaviors, capturing involved neurons requires a high induction rate with low background signals. Our ST-Cal-Light is designed to have concentrated constructs at the cell..." Please clarify.
47. p14, "Tagging neurons..." should probably be a new paragraph. Also, "can" should be "could" in this paragraph.
48. p14, "reduces expression errors or variation caused by virus injection" should perhaps be

"reduces variability associated with virus injection"

49. P34, Unclear: "All drug stock solution was prepared with 1,000x or greater," please rephrase.

50. Methods, p35, it is unclear why the authors used "Wilcoxon's signed rank", they mention it side by side with Student's t-test, as if the former is used for unpaired data and the latter for paired, but there is obviously also a paired t-test that could have been used, so this is not the right justification for using Wilcoxon's signed rank; the reason for using it was presumably because it is non-parametric so does not assume normality. And later, the unpaired non-parametric test corresponding to Wilcoxon's signed rank, i.e. the Wilcoxon-Mann-Whitney U test, is mentioned, yet it seems like this test is more closely related to the Wilcoxon rank test than the Wilcoxon rank is associated with the t-test, so it is an odd structure to the paragraph. Please tidy up this paragraph to clarify what test was used when and why.

51. Page and line numbering were missing but would have made it easier to feed back on this manuscript.

Reviewer #2:

Remarks to the Author:

As an upgrade of their original Cal-Light version, the authors successfully designed the soma-targeted Cal-Light (ST-Cal-Light) and generated a conditional ST-Cal-Light knock-in mouse to allow higher signal-to-noise ratio and an ease to be implemented according to diverse experimental needs. The cre-dependent ST-Cal-Light mouse will enable cell-type specific activity-dependent labeling, which is very desirable.

I would like the authors to address the points listed below:

1. How specific is the Cal-Light using different wave length of light? Will ambient light/yellow light cause labeling? If it is indeed highly specific to the blue light, does that mean we could NOT use it to express regular ChR2 in repetitive behavioral sessions? For example, once ChR2 is expressed, Cal-Light would activate ChR2 with blue light during labeling and thereby further induce expression of ChR2; and thus if there is background ChR2 expression, then such neurons will be further activated and labeled, resulting in non-specific labeling in unwanted cell populations.

I saw in the Methods section, the authors described pAAV:TRE-ChR2-YFP plasmid, but there is no actual result described using the ChR2 vector. This made me wonder whether ST-Cal-Light cannot be used to specifically express ChR2 in behaviorally relevant neurons. If this is the case, the authors should discuss such limitations of ST-Cal-Light.

2. Please explain how the random labeling control was done in Figure 4h. Did the random labeling match the total blue light ON time, did the authors attempt to match the number of cells labeled with opsins?

3. It would be extremely helpful if the authors could provide a table summarizing a general guideline of 1) optimal time that allows virus to express before light labeling; 2) ideal blue light on/off combinations and minimal requirement of total blue light ON time; 3) optimal wait time to allow sufficient expression and behavioral tests, but not too long such that the expression disappears.

4. Please label the conditions of the four columns of the scatter plot in Figure 1h and provide correlation coefficient and p values in the figure legend.

5. Please explain how the expression profiles are plotted in Figure 3e. Are those cell labeling density plots or just qualitative illustrations?

6. Please add mouse numbers and details in statistical comparisons in the Figure 4 legend.

7. Why are there many EGFP+ cells that do not colocalize with PV staining in Figure 5i? Please also provide percentage of colocalizations of PV or CamKII with the labeled cells in Figure 5i and Figure 5g, respectively.

Reviewer #3:

Remarks to the Author:

- 1) The authors have a previously published Cal-light system to tag active neurons. The current manuscript refines this further.
- 2) The authors claim that the Cal-light system is superior to Cfos-expression- driven activity reporter mice. This claim is grounded in a solid and logical premise. However, the authors did not directly compare the two systems to demonstrate the superiority of the Cal-light method.
- 3) One limitation of the Cal-light system is that light penetrates only a small brain tissue volume due to scattering and absorption. One would expect the number of light-tagged neurons to decline as a function of the distance from the light source. The authors must present labeling efficiency data at varying distances from the light source. This information is vital for using the Cal light system knock-in mice.
- 4) The authors call four labeling sessions (FR 8-12), with 550 seconds of blue light "weak labeling." Did the authors try to label neurons after a single session? Similarly, foot shock experiments classically involve giving a single shock, and mice freeze on subsequent exposure. Tonegawa's group has used c-fos driven tTA-expression (developed by Mayford) mice to trace the memory engram of a single shock and manipulate it. The authors need to specify whether a single shock was sufficient to label neurons or not.
- 5) Freeze experiments, please demonstrate that activating the Cal-light labeled neurons causes freezing.
- 6) Kainate acid –seizure experiments, seizures are measured best with a combination of EEG and behavior. Behavioral data alone is insufficient to support the authors' claim of seizure suppression.
- 7) The manuscript should be carefully edited to improve readability.

Point-by-point response to reviewers' comments:

Review signed by: Jesper Sjöström

Building on the elegant gene expression system (Cal-Light) that was previously reported by the same group (Lee and Kwon, Nat Biotech. 2017), Hyun and Kwon further refined the system by engineering the sensor to be soma-targeted. Briefly, this system serves as a coincidence detector of high calcium level and light activation. At high calcium, the membrane tethered sensor recruits a viral protease, which cleaves a viral linker that is only exposed by blue light activation. The cleavage process releases a transcription activator that promotes specific transgene expression in the nucleus. Since the goal of detecting high calcium lies primarily at sensing only active neurons that fire action potentials, the authors reasoned that targeting the sensor to soma would avoid off-target readout by reducing action potential-independent calcium sensing. This is particularly important in the in-vivo context of behaviours or memory-related circuits, where only the active cells within the desired time window can be precisely targeted and correlated with the task.

In cultured hippocampal neurons and cortical slices, Hyun et al. first verified that the new ST sensors are correctly enriched in the soma and enhance transgene expression up to two-fold compared to the original version. After confirming that ST sensors can be correctly induced by neuronal activity and light in vivo, the authors used the system to target halorhodopsin expression in neurons activated during four different behavioural paradigms. This showed expected behaviours to be inhibited when halorhodopsins were activated to depress neurons labelled by ST-Cal-Light, demonstrating the system effectiveness and utility in vivo. To partially circumvent the need to introduce multiple components of the ST-Cal Light system, the author generated a transgenic mouse line that expresses the sensor in a Cre-recombinase dependent manner. The mouse line was then verified for the ST-Cal Light system after crossing with two commonly used Cre lines.

The design of the new sensor with the soma-targeted motif addition is a potentially very important modification. The proof-of-principle behavioural experiments are well designed and performed. The results are convincing, with good controls for most of the experiments. Figures are of high-quality, although at times a bit compact, and the text is succinctly written, although more details in the Methods section would help and more explanatory text would help to clarify. The ST-Cal-Light mouse line will likely be a valuable contribution to the field. One caveat is possibly the leakiness of expression, the authors could be more upfront about this. After addressing these comments, we believe the present manuscript should be highly suitable for Nature Communications.

We thank Dr. Jesper Sjöström for constructive comments. We are pleased that the reviewer consider our study particularly important and suitable for publication in *Nature Communications* should we address suggested concerns. Accordingly, we have performed a significant number of new experiments and analyses, presented all data values, clarification of the technique, described in detail below, which we hope the reviewer will find that our revised manuscript is strengthened and suitable for publication.

MAJOR POINTS:

1) Fig. 1h and 1k:

a. Correlation coefficient and p values should be presented to support the correlation mentioned in the main text.

We carefully re-analyzed the data and concluded that there is no 'positive correlation between green and red fluorescence'. Based on this result, we removed the sentence "overall tendency of positive correlation between green and red fluorescence was observed" from the main text. We added the correlation coefficient and p values in the supplementary table 1.

Table 1 Pearson Correlation for Cultured Neuron (related to Fig. 1h)

	Dark-TTX	Dark-BIC	Blue-TTX	Blue-Bic
ST-Kv2.1	0.10346	0.21993	0.13303	0.11002
ST-KA2	-0.01705	0.29293	0.22200	0.01199

Table 2 P-value for Cultured Neuron (related to Fig. 1h)

	Dark-TTX	Dark-BIC	Blue-TTX	Blue-Bic
ST-Kv2.1	0.07859	0.00007	0.03947	0.06602
ST-KA2	0.77632	0.00000	0.00024	0.83612

Table 3 Pearson Correlation for Slice Culture (related to Fig. 1k)

	Dark	Light Only	Stim Only	Light + Stim
ST-Kv2.1	0.00042	-0.05128	-0.02235	0.00738
ST-KA2	0.24405	0.15772	0.48039	0.36798

Table 2 P-value for Slice Culture (related to Fig. 1k)

	Dark	Light Only	Stim Only	Light + Stim
ST-Kv2.1	0.99591	0.40129	0.75342	0.90217
ST-KA2	0.00004	0.00664	0.00000	0.00000

b. How were the cutoffs (dotted lines) defined in the graph for B, G, Y and R quadrants? The y-axis cutoff actually changed from ~70 a.u. in Fig.1h to 100 a.u. Fig.1k.

Thank the reviewer for the comment. We set the y-axis cutoff by scaling to the value of red fluorescence. For example, the maximum values of red fluorescence from the dissociated culture neurons and the organotypic slice cultures were 255 a.u. and 365 a.u., respectively. If we scaled the y-axis cutoff in proportion to these red fluorescence max values, then the cutoffs become 70 a.u. and 100 a.u, respectively. The following is how we calculated; 255: x = 365: 100 where the x value is approximately 70 a.u.

c. Presumably raw signal values are used for these plots, were the imaging conditions consistent across groups and experiment days? In our experience, virus injection does not always result in the same expression level between experiments. One quick way to avoid spurious patterns is to colour-code the datapoints by experiment days and ensure they don't form clusters by dates.

We thank the reviewer for pointing out this issue. In our hand, results obtained from the separate experiments were similar and the imaging procedures were performed consistently. We plotted data points with different colors by separate experimental days as follows. As seen in these figures, data do not form discrete clusters in the distributed plot. This new data set was included in the supplementary fig. 1.

2) There appears to be a considerable proportion of cells with transgene expression but with no TdTomato reporter expression, in particularly in the Light + Activity condition and in vivo. This is suggestive of the M13-TEV-C-P2A-Tdtomato was not successfully transduced into these cells, and that the transgenes were expressed non-specifically in the absence of the TEV-C protease – a key component for Cal-Light to work correctly. For example, Fig. 2e, 3c, 4b, 4i, 5f, 5g and 5i. A couple of concerns arise here:

a. How were the G/R ratios treated in these instances? (Also relevant to Fig. 1g and 1j) The virtual absence of R can result in the artefact of apparent large increases in the G/R ratio. Was a R signal threshold applied to select certain population of cells for quantification? How was this threshold determined? Without a good procedure, the G/R ratio can be skewed by using cells with (background) R signal coming largely from the neurites of other Tdtomato expressing cell.

We thank the reviewer for his thoughtful and constructive comments. As pointed out, we observed some cases with low tdTomato expression but strong green fluorescence. However, we believe that this is not due to the failure of protein transduction into cells nor the nonspecific EGFP expression in the absence of the TEV-C protease. That is based on several key evidence regarding this issue. We performed many control experiments in the previous papers (Lee et al, 2017, Nature Methods; Lee et al, 2017, Nature Biotechnology) showing evidence for this.

The first evidence was that when we transfected mutant TEV-C components, almost zero reporter gene expression was observed (see the figure below, *Lee et al, 2017, Nature Methods, Supplementary Figure 8*). It was reported as follows;

“We further tested spontaneous association of the split TEV system in the iTango2 system. The original split TEV system is reportedly leaky. This means that restoration of TEV protease activity is partially independent of external signals such as pharmacological drugs or light. This “leakiness” may have originated from the intrinsic affinity of C- and N-terminal fragments or difficulty in separation once they associate. This problem was alleviated by creating a deletion mutant of TEV-C. We used this truncated form of TEV-C for our iTango system, but we further deleted its C-terminus to minimize background levels. However, further deletion of three amino acids completely eliminated protease function implying that the current truncated version of TEV-C is the smallest possible fragment to lower background level without losing intact protease activity (Supplementary Fig. 8)”.

This experiment tested a truncated version of TEV-C functions to minimize the leakiness of spontaneous gene expression. **However, it also indicated that the other component (tTA containing construct) could not make gene expression by itself or even if that is possible, the level is extremely low.** Therefore, the partner construct, TEV-C, must be present in order to make substantial light-dependent gene expression.

Secondly, we also addressed this issue in the original Cal-Light paper (*Supplementary figure 5 of Lee et al, 2017, Nature Biotechnology*). Even if we expressed TEV-C-P2A-TdTomato, but M13 was deleted (*deM13-TEV-C-P2A-TdTomato*), then almost zero reporter gene expression was made. Red fluorescence distribution was all same across conditions, but only green fluorescence was shifted depending on M13. These results again strongly indicate that very bright EGFP signal was “mostly” induced by light and calcium dependent manner, not by adjusted red fluorescence level.

For analysis, we did not pick cells with low R background signals. We included all neurons at each experimental condition and always R level was analyzed in the consistent manner without adjusting background intensity. As exemplified in Fig. 2e, major changes were only green signals in “Light + Activity” conditions. R signals were similar across all conditions. To avoid any thresholding effect set by R signal, we analyzed only green signals. As expected, the results were similar to those shown in G/R (Supplementary Fig. 2). Altogether, we would like to say that our main conclusion was not made by inappropriate analysis of R fluorescence nor skewed by a few strong G fluorescence.

b. Can the authors speculate why this is happening? It is highly advisable to check the detailed subcellular localization of the three components of ST-Cal-Light. One possibility is that some of the transmembrane sensors are mistargeted to the nuclear membrane instead of the cell membrane, allowing the TetO sequence in the nucleus to interact with tTA that is intact in the sensor, and in the absence of the TEV-C protease. While we appreciate an immediate remedy to the problem may not be possible, the caveats and speculations should be frankly noted in the discussion to help other users.

We agree that this is an important point and we thought about the possible reasons. Although we do not know exactly what caused this phenomenon, one thing clear is that EGFP reporter gene expression should be made when TEV-C-P2A-TdTomato construct is present. What it means is that TdTomato component should be present in the green positive neurons. If tTA was mistargeted to the nuclear membrane allowing the reporter gene expression in the absence of the TEV-C, we should be able to detect strong EGFP signals without TEV-C. However, we did not see these results by experiments described above. One possible scenario is that in some neurons, a portion of tTA was cleaved out via normal Cal-Light processes (TEV-C and TEV-N reconstitution and TEV sequence exposure by light), but gene expression became abnormally high or uncontrolled for some reason at the gene transcription level. It is also possible that protein aggregation of TEV-C and TEV-N is made in subcellular region, which may cause excessive gene transcription. We described this issue in the discussion section.

c. To estimate the leakiness of the ST-Cal-Light, quantification should be performed and presented in the manuscript to account for the percentage of cells with G signal but almost no R signal.

For the reason we described above, considering the “high G/low R” portion the leakiness of ST-Cal-Light system does not seem to be logically correct. We believe that the population of cells with G signal with low R signal should be all included in the data analysis and considered the part of normal Cal-Light actions.

d. G/R ratio quantifications are missing for the the Cal-Hom: EMX-Cre and Cal-Hom: PV-Cre experiments.

We quantified G/R ratio for the Cal-Hom: EMX-Cre and Cal-Hom: PV-Cre experiments as requested. The new data set was included in Fig. 5j.

3) To improve the impact of the ST-Cal-Light system, it would be important to compare how the new system with soma-targeting outperforms the old Cal-Light.

a. A comparison between Fig. 2 and their previous study (Lee and Kwon, Nat Biotech. 2017) seem to show very similar readouts with minimal improvement – G/R ratio, Lever press per min, Cumulative lever press.

We thank the reviewer for the comment. We carefully compared the performance of the OG-Cal vs. ST-Cal-Light and found that there were significant differences in the inter-reward interval and the pressing-licking matching ratio between them (see supplementary fig. 15 in Lee and Kwon, Nat Biotech. 2017). “Lever press per min” and “Cumulative lever press” indicate learning capability of mice. They are not readouts representing the Cal-Light efficacy. General learning curve will be similar whether we inject the old or new Cal-Light viruses.

In order to demonstrate the outperformance of the ST-Cal-Light, we tested “mild label” procedure in this manuscript. Because only “Full label” protocol was used in the previous paper, we also used the same protocol by using the ST-KA2-Cal, but the “full label” protocol seemed to be close to the saturation level. Better efficiency or inducibility was difficult to be revealed by “full label” protocol. So, we reduced the labeling time by using the ST-Cal-Light and confirmed that labeling was still sufficient to label learning-specific neurons.

More importantly, we further examined the labeling performance of ST-KA2-Cal caused by single-session lever press-induced gene expression (only during the FR12 session). Such reduced labeling time also resulted in robust gene expression. This new data set was added in the Fig. 2f).

b. It is possible that the above longer term labelling paradigm does not fully exploit the faster labelling capability of ST-Cal-Light. The manuscript would benefit from comparing old vs. new system in the fear conditioning paradigm (short labelling time), in which the total blue light exposure may reveal the improved efficiency.

Thanks for the suggestion. We directly compared the efficiency of old and new systems in the fear conditioning paradigm. In figure 3a-3b, we shined the 5-second duration of blue light (x 3 times) coincidentally with a foot shock. After labeling with the ST-Cal-Light, inhibiting the activity of labeled neurons was sufficient to inhibit freezing responses during the retrieval phase. However, the same labeling protocol using the OG-Cal-Light did not reduce the freezing behavior by 589 nm light (Supplementary Fig. 4). These results demonstrate the improved efficacy of the ST-Cal-Light and the sufficiency of behavioral changes with short labeling time. The new result is added in the supplementary Fig. 4.

4) The manuscript is not always as clear as it could be. Although Fig 1a clarifies, it wasn't immediately clear from the text how Cal-Light works and how light is involved, please describe on page 5, top. In general, the manuscript can be strengthened by including more details on procedures and analyses in the Methods section. It was not immediately easy to understand what was done with

each experiment or figure unless one carefully reads the main text, figure captions, and methods section together. The light is an important component for the temporal resolution so it would be helpful to elaborate on that in the abstract. In general, figure captions should state what the figure means or what the reader should conclude, not what is in the figure, e.g. Fig.

ST-KA2-Cal-Light (Single shock; 1x)

OG-Cal-Light (3x)

G/R ratio comparison

2 "Confocal images showing EGFP expression at each condition" could be rephrased, for clarity's sake. Same thing with "Average of lever press number per minute under condition" and "Cumulative lever press number plotted against time," because this is not very informative, as the reader can already see this from the figure.

We thank the reviewer for several suggestions to improve visibility of our manuscript. Because how Cal-Light works and how light is involved were well described in the first Cal-Light paper, we just briefly explained again how Cal-Light work in this paper (added in the introduction). In addition, we also explained about the optimal conditions and things to consider when ST-Cal-Light is used. This part was added in the discussion section.

We rephrased "Confocal images showing EGFP expression at each condition" to "*When cell labeling is finished, brain was fixed and the degree of gene expression was quantified by confocal imaging.*" in Fig. 2. We also rephrased from "Average of lever press number per minute under condition" and "Cumulative lever press number plotted against time" to "*The total lever pressing number were compared before and after 589 nm light, and the following day of inhibition test. Both mild label and full label conditions were plotted.*" and "*The number of lever press was plotted over time to demonstrate how fast animals reach the goal. Note that more time was required to reach 250 lever presses when 589 nm light is turned on after labeling.*", respectively.

MINOR POINTS:

1) Fig. 1i is not referred to in the main text
We mentioned Fig. 1i in the corresponding text.

2) Fig. 2l- Significance asterisks missing between 589 OFF and 589 ON Full label conditions?
We added asterisks on the figure to demonstrate the significance.

3) Fig. 2j- Meanings of the colour bars underneath the FR-12 label are not clear, please define in figure caption

We explained the meanings of the color bars in the figure legend. "*inset: blue horizontal bar underneath the FR-12 label indicates the last day of training with labeling in the presence of blue light. The yellow horizontal bar indicates the probe-test day in the presence of yellow light throughout the session (2 sec ON, 1 sec OFF). The bar half-filled with blue indicates the following day in the absence of yellow light but labeled with blue light during training.*"

4) Fig. 4h- The tiny orange strand in black colour scheme for the "2nd KA inj" condition is not apparent. The orange colour implies the 589 laser is ON when it is not.

We apologize for the confusion. We removed the tiny orange strand from the second column bar graph.

5) Fig. 4i- Top row labels are colour-coded but not at elsewhere in the manuscript, the authors may want to standardize them to improve consistency.

All labels are color-coded in a consistent manner.

6. The figure captions could be improved with more detail. For example, in figure 1j and figure

2g, it would be useful for the readers if data values were included (unless if it goes against journal guidelines). A good example would be the caption of figure 1g. Another example is the caption for figure 3i, could it be clarified whether each data point corresponds to one mouse?

We added data values in the figure legend. We also clarified that each data point corresponds to one mouse.

7. It was mentioned that mice moved to FR sessions of the lever pressing motor learning task after obtaining 15 successful rewards in the CRF sessions. It is unclear to me in Figure 2b how this happens in Day 5 because the illustration tells me that it would be after 25 rewards, so perhaps some clarification here would be helpful.

15 rewards meant the number of rewards only during the CRF 15Rs at day 5, not the sum of rewards until day 5. We clarified the sentence as follows. *“During the CRF 5Rs and CRF 10Rs, mice received five rewards and ten rewards at each training day, respectively.” When mice finished getting fifteen rewards (CRF 15Rs) at the training day 5, they were moved to the fixed ratio (FR) schedule, which consists of FR-2*”. This new sentence is added in the main text.

8. Figure 2d was not cited in the main text.

It is now correctly mentioned.

9. Figure 3k is cited before 3j in the main text.

We fixed it. The figure number appears in order.

10. Introduction paragraph 2: what are some examples of the high cognitive behaviors? Consider rephrasing as higher-order cognitive behaviors.

Thank the reviewer for suggestion. We rephrased it as higher-order cognitive behaviors.

11. Introduction paragraph 3: Not sure how non-specific signals would accumulate over time while waiting for viral expression. There would be no light, right? Is it from construct remaining Ca-bound?

Yes, there will be no light in the mouse brain. Nevertheless, we have to consider potential spontaneously arising non-specific signals because no protein interaction or gene expression are not controlled in 100% yes or no manner. In vitro studies showed that there are some gene expression even in the dark condition although the level is extremely low. Such signals may accumulate more in a longer period of time. Reducing even this portion will be important to increase the signal-to-noise ratio.

12. Results: Since you spelled out kainate receptor 2 (KA2), it might be useful to also indicate Kv2.1 as a potassium channel.

We added voltage-dependent potassium channel.

13. Results end of paragraph 1: what is the "other partner"?

The other partner was just explained above the mentioned sentence in the end of paragraph 1. It is either *Myc-TM-KA2* or *Kv2.1 motif-CaM-TEV-N-AsLOV2-TEVseq*.

14. Perhaps the brightness/contrast in Fig 1B bottom panels can be adjusted? Right now, it is

hard to tell the OG-Cal expression in dendrites

We adjusted the brightness/contrast to visualize GC-Cal expression better.

15. Figure 1h - Since ST-Kv2.1 and ST-KA2 plots are already indicated in the figure, they could perhaps be color-coded in the same way as Figure 1g - according to experimental condition.

We considered changing this figure color-coded, but this change made the figure has too many colors. For this reason, we decided to keep the current color format.

16. Figure 2b - unclear what 5Rs, 10Rs, 15Rs mean

They mean that mice receive five (5Rs), ten (10Rs), and fifteen rewards (15Rs), respectively. We added this information in the figure legend.

17. While figure 2c is nice, I don't feel like it accurately represents the 5 sec ON/25 sec OFF cycle). Consider revising or including it in fig 2b?

We included this information in the Fig. 2b.

18. How is NpHR expressed? This method is very interesting so I think you should elaborate more in the text. Also, it is surprising that NpHR only needed 2 days to be sufficiently expressed to show a behavioural outcome.

To understand the time course of reporter gene expression, we previously monitored when light-induced reporter gene expression begins, when it reaches the peak, and how long it stays in vitro and in vivo (See supplementary figure 11 of Lee et al, 2017, Nature Methods). Although this experiment was performed with the iTango2 technique, the light-sensitive module is the

identical with the ST-Cal-Light technique, so the protein life cycle of expressed reporter will be the same. We found that the maximum level was observed about 48 hours after the blue light illumination.

Testing behavioral causality 2 days after labeling is not so surprising. Other active cell tagging methods (e.g., c-fos based cell labeling) also showed behavioral control after allowing 2 day expression of reporter protein (Liu X et al, 2012, Nature) such as channelrhodopsin and halorhodopsin.

19. Figure 2j, k: What about off-target expression of NpHR? Is there a no label condition (no blue light)?

To measure the off-target expression of NpHR, we performed additional experiments comparing no light, 589 nm light, and blue light with different fiber optic lengths. As seen in the figure below, if blue light was not delivered, no EGFP reporter was expressed. This figure is added in the Supplementary Fig. 3.

20. Page 8 "reflecting the erasure of memory linking level pressing to rewards" - I don't think this is true. Since you are inhibiting activity in M1, it is possible that the motor function was inhibited but the memory is not erased. I also don't think the memory was erased, because the day after 589 ON, the behaviour was restored. So you are inhibiting the behaviour (most likely scenario) or preventing the recall of the trained behaviour. Please clarify.

Thank the reviewer for this comment. When we analyzed the lever press-licking matching ratio in Fig. 2n, we found that even if mice pressed the lever, the probability of going to water reward magazine was significantly dropped. This result suggests that the lever pressing motor action was still normally functioning, but the memory that the lever press was associated with the water reward was inhibited. I agree that this behavioral change may not indicate the complete erasure of memory linking, but rather preventing the recall of the learned behavior. We changed the sentence accordingly.

21. Page 9, were the foot shocks paired with an unconditioned stimulus? What was the procedure during the retrieval phase?

We apologize for omitting a description of the retrieval phase in our method section. The full revised description is as follows: on day 1, mice underwent the habituation phase for 10 min in a neutral context (i.e., chamber). The mice were then subject to the fear acquisition phase for 5 min, in which the context was paired with three foot-shocks (for each, 2 s and 0.7 mA) delivered at 120 s, 180 s, and 240 s, respectively. Two days later, for the retrieval of contextual

fear memory, the mice were re-exposed to the conditioning context and freezing responses were measured for 5 min. This is added in the method section.

22. Figure 4g - consider changing the light grey line to a darker color

To have better contrast, we kept the same color.

23. Figure 5i - it looks like cells are not co-expressing EGFP and PV? Is there further cell-counting data?

We quantified the number of PV-positive cells expressing EGFP and the number of EGFP+ cells colocalized with PV signals. These results are added in the Supplementary Fig. 6.

24. Page 6, "To verify whether similar results are also obtained in brain slices", please specify that these are not acute slices.

We specified to "slice cultures".

25. Page 7 and p9, "shined" should be "shone"

This is now fixed.

26. Page 7, "To compare with a weak labeling...", please clarify the rationale for this comparison.

To clarify the rationale, we changed the sentence as follows; "To test whether ST-KA2 enables efficient labeling with shorter period of blue light illumination time,..."

27. Page 7, "... to match the same conditions in which ...", please clarify which conditions.

The condition was already described in a few sentences before this sentence and also well explained in the previous paper which we cited as a reference.

28. Page 8, "NpHR" in the text but "eNpHR" in the figure, please be consistent.

We fixed it in the figure.

29. p8, typo, "the number of lever pressES"

We fixed this typo.

30. p8, "Once mice finished FR12 training, 589 nm..." Does 589 nm light not induce Cal-Light? Please show.

As shown in Q19, we performed labeling experiments using 589 nm light. As demonstrated in the supplementary figure 3, EGFP reporter gene expression was not induced by 589 nm light.

31. p8, "not due to the tissue damage", please remove "the"

We deleted "the"

32. p8, "reporter was used as a negative control," (comma missing)

We added "," as the reviewer pointed out.

33. p8, "ST-KA2, M13-TEV-C, and TetO-NpHR-EYFP", please explain in text what each one does, for clarity.

What each construct does was described in detail in our previous paper (*Lee et al, 2017, Nature Biotechnology*). Because these basic molecular principles were already reported in the original Cal-Light, here we added brief description about how Cal-Light system works. This information is added in the introduction as follows.

"Cal-Light system requires two separate synthetic proteins containing C- and N-terminus of tobacco etch virus (TEV), which is fused to calmodulin (CaM) and M13 protein, respectively. When Ca²⁺ level goes up, CaM and M13 proteins bind to each other causing the restoration of TEV protease functions. Upon blue light illumination, TEV cleavage sequence (TEVseq) is exposed to the cytosol, recognized by TEV protease, and allows tethered tetracycline-controlled transcription activator (tTA) to go to the nucleus initiating gene expression."

34. p10, "It is generally believed that epileptic seizure is caused by unobstructed", unclear what "unobstructed" means here, please rephrase.

We changed "unobstructed" to "uncontrolled" to eliminate potential misleading.

35. p10, "gene expression was only present in the group ...", Can we get a cell count too, pls?

Each circle indicates the cell. We provided all cell numbers in the figure legend.

36. p10, it might be useful to cite and very briefly discuss other optogenetics approaches in epilepsy research, to contextualize the pros and cons of the present approach.

Thank the reviewer the comment. We briefly described other optogenetics approaches in epileptic seizure research and what makes the present approach different from them in the main text. *"Optogenetic approaches to control epileptic seizure has been demonstrated in the past decade, but the controlling target was a specific brain area or a cell type (Tung JK, 2016, Brain Stimulations; Krook-Magnuson E et al, 2013, Nature Communications; Bui AD et al, 2018, Science). These approaches affect the activity of the entire target brain area or cell population, which include both disease-related and normally functioning neurons. Interrogating only seizure-engaged neurons would have benefit by minimizing the side effect but resulting in the similar amelioration of seizure symptoms"*.

37. p11, please remove "of the chromosome"

We deleted "of the chromosome" as suggested.

38. p11, "A LoxP flanked neo cassette was placed in the upstream of target gene" should be "A LoxP flanked neo cassette was placed upstream of the target gene"

We fixed it.

39. p11, typo "blue lights"

We fixed it.

40. p12, "into the layer2/3", please remove "the"

We removed "the"

41. p12, several typos, see uppercase here: "tdTomato signal WAS driven by CAG promotEr, so its expression was not restricted TO specific cell types, but EGFP expression was induced in excitatory neurons because Cre expression WAS limited TO neocortical excitatory neurons under THE Emx1 promoter"

We fixed all.

42. p12, poor grammar, also unclear: "Thus, conditional ST-Cal-Light mice specify active neurons from a distinct cell type."

We changed the sentence as follows; *"Thus, conditional ST-Cal-Light mice enable to target active neurons out of the genetically defined cell type"*

43. p13, generally unclear: "because somatic action potentials are typically final output signals of neurons, their inhibition has a more prominent effect on behaviors" Should 'typically' be 'always'? Etc

We changed the sentence as follows; *"because somatic action potentials are always the final output signals of neurons, their inhibition negates integrative synaptic potentials resulting in prominent effects on behaviors"*

44. p13, "while less active neurons DO not"

We fixed it as suggested.

45. p13, "Labeling efficiency ... ", this sentence is unclear. Also, it should be in past tense, because this is a reported finding, right?

This sentence explains the general features of the Cal-Light technique when other people consider experimental plans. It is not just finding only applied to the result reported in this manuscript. Therefore, we used the present tense. To further clarify it, we added an additional separate paragraph describing the optimal conditions when using the ST-Cal-Light technique in the discussion as follows.

"The optimal condition for the best labeling would be different in case by case. Dependence on Ca²⁺ and light is the basic operating system of the ST-Cal-Light. What it means is that any factor that affects Ca²⁺ levels and how it is matched to the blue light protocol will be critical. Because different types of cells have different intrinsic properties (e.g., resting membrane potentials, ion channel distributions, input resistance), it is difficult to make a guideline that universally applies to all cells. Nevertheless, several important rules should be considered.

First, the expression level of two ST-Cal-Light components is important. If they are expressed in a similar amount (e.g., 1:1 ratio), the labeling efficiency increases. If the M13 containing

construct is much higher or lower compared to the CaM-containing construct, then labeling efficiency decreases (Lee et al, 2017). That is because these two constructs work together in order to begin gene expression. Second, labeling efficiency increases if cells fire as a short burst with a high frequency. Even if the number of action potentials and blue light exposure time are the same, the higher reporter gene expression is made if firing occurs as a short burst (Lee et al, 2017). Third, the number of blue light repeats are more important than the total duration of blue light. The behind logic is that the onset timing of AsLOV structural modification by blue light is very fast, but the restoration to the original structure takes tens of seconds. Due to such light responsiveness, a brief light pulse (1~2 seconds) followed by some interval (~30 seconds), and then repeating the same protocol will maximize the labeling efficiency.”

46. p13, these three sentences are generally unclear and hard to follow: "In case of transient behaviors, capturing involved neurons requires a high induction rate with low background signals. Our ST-Cal-Light is designed to have concentrated constructs at the cell..." Please clarify.

We tried to explain this part as simply as possible in the discussion section because this part was already explained quite in detail in the introduction. To have better understanding, we modified the sentences as follows; *"In the case of transient behaviors, capturing involved neurons requires a high induction rate with low background signals. Our ST-Cal-Light is designed to cause the Cal-Light protein expression condensed in the cell soma, the chance to cause protein-protein interaction (two Cal-Light components) increases upon each blue light exposure. Therefore, the sufficient level of reporter gene expression can be obtained via a small number of light repeats"*

47. p14, "Tagging neurons..." should probably be a new paragraph. Also, "can" should be "could" in this paragraph.

We modified the text as suggested.

48. p14, "reduces expression errors or variation caused by virus injection" should perhaps be "reduces variability associated with virus injection"

We changed this phrase as suggested.

49. P34, Unclear: "All drug stock solution was prepared with 1,000x or greater," please rephrase.

This sentence is rephrased as follows; *"They were prepared as stock solution in high concentration (1,000x or greater) and the final concentration was met via dilution"*

50. Methods, p35, it is unclear why the authors used "Wilcoxon's signed rank", they mention it side by side with Student's t-test, as if the former is used for unpaired data and the latter for paired, but there is obviously also a paired t-test that could have been used, so this is not the right justification for using Wilcoxon's signed rank; the reason for using it was presumably because it is non-parametric so does not assume normality. And later, the unpaired non-parametric test corresponding to Wilcoxon's signed rank, i.e. the Wilcoxon-Mann-Whitney U test, is mentioned, yet it seems like this test is more closely related to the Wilcoxon rank test than the Wilcoxon rank is associated with the t-test, so it is an odd structure to the paragraph. Please tidy up this paragraph to clarify what test was used when and why.

We apologize for the unclear description in statistics. For assessing normality of data

distribution, we performed the Kolmogorov-Smirnov test. For comparison between two independent samples, we applied two-tailed Student's *t* test (parametric) or Mann-Whitney *U* test (non-parametric) after testing normality. For comparison between two conditions from same samples, paired *t* test was applied after confirming normal distribution of data. For comparison of data from multiple samples that shows normal distribution, ANOVA was applied. In particular, when the data was obtained in several conditions for multiple times, two-way RM ANOVA was performed. Bonferroni correction was used for correction of post-hoc multiple comparisons.

51. Page and line numbering were missing but would have made it easier to feed back on this manuscript.

We added line number and pages.

Reviewer #2 (Remarks to the Author):

As an upgrade of their original Cal-Light version, the authors successfully designed the soma-targeted Cal-Light (ST-Cal-Light) and generated a conditional ST-Cal-Light knock-in mouse to allow higher signal-to-noise ratio and an ease to be implemented according to diverse experimental needs. The cre-dependent ST-Cal-Light mouse will enable cell-type specific activity-dependent labeling, which is very desirable.

We are grateful to the reviewer for prompting us to perform important control experiments, analysis, and suggestions that have greatly enhanced and strengthened our work. We appreciate that the reviewer takes considerable time and effort, and we do not take this for granted. Thank you.

I would like the authors to address the points listed below:

1. How specific is the Cal-Light using different wave length of light? Will ambient light/yellow light cause labeling? If it is indeed highly specific to the blue light, does that mean we could NOT use it to express regular ChR2 in repetitive behavioral sessions? For example, once ChR2 is expressed, Cal-Light would activate ChR2 with blue light during labeling and thereby further induce expression of ChR2; and thus if there is background ChR2 expression, then such neurons will be further activated and labeled, resulting in non-specific labeling in unwanted cell populations.

We thank the reviewer for the comment. We do not see much gene expression when cells were labeled with general ambient room light (no blue light) or yellow light (see below). Ambient room light may cause some labeling when culture neurons are not properly handled without light protection, but mouse brain was dark enough not to cause cell labeling.

Even if the Cal-Light is specific to blue light, ChR2 can still be used together as a reporter. However, in this case, repetitive, multiple rounds of ChR2 activation should be avoided. For example, when ChR2 reporter is used, the first blue light exposure during the labeling process will begin to express ChR2. Then, a few days later, when ChR2 is expressed in the labeled neurons, the second blue light exposure will cause activation of ChR2, which will show behavioral causality. During this period, the second labeling process will begin. But, if the experiment is terminated at this step, then the effect of secondary labeling process will not be an issue any longer. So, we do not recommend using the ChR2 reporter when behavioral sessions are planned to run repetitively for multiple days.

For the activation purpose, we recommend using a red-shifted reporter (e.g., ChrimsonR)

rather than ChR2. Labeling by blue light and reactivation by yellow light are now shown in fear conditioning experiments (Fig. 3).

I saw in the Methods session, the authors described pAAV:TRE-ChR2-YFP plasmid, but there is no actual result described using the ChR2 vector. This made me wonder whether ST-Cal-Light cannot be used to specifically express ChR2 in behaviorally relevant neurons. If this is the case, the authors should discuss such limitations of ST-Cal-Light.

I apologize that our manuscript was not carefully written regarding this comment. We removed pAAV:TRE-ChR2-YFP plasmid. As explained above, the ST-Cal-Light can be used to express ChR2, but using ChrimsonR reporter is a better option because its excitation wavelength is not overlapped with blue light.

or have we used pAAV:TRE-ChR2-YFP for the sociability experiment?

We have not used TRE-ChR2-YFP, but instead, we used TRE-ChrimsonR-EYFP for the contextual fear memory task.

2. Please explain how the random labeling control was done in Figure 4h. Did the random labeling match the total blue light ON time, did the authors attempt to match the number of cells labeled with opsins?

The random labeling used for Fig. 4h did not match the total blue light ON time with epileptic seizure labeling procedure. We delivered blue light for 5 seconds, 3 times.

To match the total blue light repeats and duration, we performed additional experiments. In this experiment, we injected the same set of viruses that were used for seizure experiments and trained the animal for lever pressing. We shone blue light, 3 sec ON/2 sec OFF, for 5 min during the lever press training (from CRF 15Rs to FR12). The total duration of blue light was 30 min, which was the same duration used for the KA-induced seizure experiments. With this labeling procedure, we observed robust labeling in granule cells, CA1, CA3 neurons, and mossy cells in the hippocampus (Supplementary Fig. 5). However, when we shone 589 nm light after the second KA injection, the epileptic seizure score maintained high (Supplementary Fig. 5). This paragraph is now added in the main text.

3. It would be extremely helpful if the authors could provide a table summarizing a general guideline of 1) optimal time that allows virus to express before light labeling; 2) ideal blue light

on/off combinations and minimal requirement of total blue light ON time; 3) optimal wait time to allow sufficient expression and behavioral tests, but not too long such that the expression disappears.

Thank the reviewer for suggestion. The optimal conditions were discussed in the original Cal-Light paper in detail (see figures below taken from Lee et al, 2017). So, we will not show the figure again in this paper, but we added description about important considerations in the discussion section as follows.

“The optimal condition for the best labeling would be different in case by case. Dependence on Ca^{2+} and light is the basic operating system of the ST-Cal-Light. What it means is that any factor that affects Ca^{2+} levels and how it is matched to the blue light protocol will be critical. Because different types of cells have different intrinsic properties (e.g., resting membrane potentials, ion channel distributions, input resistance), it is difficult to make a guideline that universally applies to all cells. Nevertheless, several important rules should be considered.

First, the expression level of two ST-Cal-Light components is important. If they are expressed in a similar amount (e.g., 1:1 ratio), the labeling efficiency increases. If the M13 containing construct is much higher or lower compared to the CaM-containing construct, then labeling efficiency decreases (Lee et al, 2017). That is because these two constructs work together in order to begin gene expression. Second, labeling efficiency increases if cells fire as a short burst with a high frequency. Even if the number of action potentials and blue light exposure time are the same, the higher reporter gene expression is made if firing occurs as a short burst (Lee et al, 2017). Third, the number of blue light repeats are more important than the total duration of blue light. The behind logic is that the onset timing of AsLOV structural modification by blue light is very fast, but the restoration to the original structure takes tens of seconds. Due to such light responsiveness, a brief light pulse (1~2 seconds) followed by some interval (~30 seconds), and then repeating the same protocol will maximize the labeling efficiency.”

4. Please label the conditions of the four columns of the scatter plot in Figure 1h and provide

correlation coefficient and p values in the figure legend.

We added the conditions and also added correlation coefficient and p values in the supplementary table 1.

5. Please explain how the expression profiles are plotted in Figure 3e. Are those cell labeling density plots or just qualitative illustrations?

Figure 3e illustrates a series of coronal schematics showing the extents of AAV expression (red) and activity-dependent labeling (green). The extents were traced based on fluorescent images taken at a low magnification (2.5x) for each animal (n = 5). Darkness represents coincidence from different animals. This explanation is added in the figure legend.

6. Please add mouse numbers and details in statistical comparisons in the Figure 4 legend.

We added mouse numbers and statistical comparisons in the Fig. 4.

7. Why are there many EGFP+ cells that do not colocalize with PV staining in Figure 5i? Please also provide percentage of colocalizations of PV or CamKII with the labeled cells in Figure 5i and Figure 5g, respectively.

We analyzed the percentage of EGFP+ neurons out of CaMKII+ neurons or PV+ neurons. We also analyzed the percentage of EGFP+ neurons colocalizing with PV or CaMKII. About half of neurons out of CaMKII+ or PV+ neurons were GFP+ (3 sec ON/2 sec OFF, 30 min in a home cage). The majority of EGFP+ neurons (86% and 82%) were also positive to CaMKII or PV antibody staining. This quantification is plotted in the Supplementary Figure 6.

Reviewer #3 (Remarks to the Author):

We thank the reviewer for suggesting valuable comments and suggestions for new experiments, which are critical for improving the quality of the manuscript. We performed many new experiments to address remaining concerns. We hope that the reviewer now think that the manuscript has been significantly improved.

1) The authors have a previously published Cal-light system to tag active neurons. The current manuscript refines this further.

2) The authors claim that the Cal-light system is superior to Cfos-expression- driven activity reporter mice. This claim is grounded in a solid and logical premise. However, the authors did not directly compare the two systems to demonstrate the superiority of the Cal-light method.

We appreciate for this criticism. We think that direct comparison between the ST-Cal-Light or

OG-Cal-Light with c-fos-mediated active cell labeling is not very meaningful, because these two systems are fundamentally different in terms of working principles. We did not claim that the Cal-Light system is “always” or “generally” superior to the c-fos-driven gene expression. The advantages of the Cal-Light dependent labeling are “high temporal resolution” and more “action-potential” dependent. Labeling and controlling engram neurons via c-fos driven tTA expression may be superior method when used in appropriate behavioral paradigms such as fear memory by foot shock. We are reporting in this paper that the ST-Cal-Light has a light switch, which enables to choose “when to label” during the behavior. Due to this light switch, labeling intensity may not be as strong as c-fos driven gene expression system, but labeling period becomes more timely resolvable. When we tested the ST-Cal-Light in single foot shock, the labeling was not sufficient to prevent freezing behavior when the labeled neuronal activity was inhibited. This result should not be interpreted by the inferiority of the ST-Cal-Light. We think that having an additional tool to label active neuronal population with a different working logic offers multiple options when other investigators design their experiments.

3) One limitation of the Cal-light system is that light penetrates only a small brain tissue volume due to scattering and absorption. One would expect the number of light-tagged neurons to decline as a function of the distance from the light source. The authors must present labeling efficiency data at varying distances from the light source. This information is vital for using the Cal light system knock-in mice.

To test the efficacy of cell labeling depending on the distance from the light source, we tested two different lengths of optic fibers. In one case, the tip of optic fiber was positioned 0.4 mm above the virus injection site and the other case has the optic fiber located 0.8 mm above the virus injection site. Blue light was delivered to the mPFC for 30 min (2 sec ON/1 sec OFF) in both conditions. Interestingly, we found that the labeling efficiencies were similar in terms of G/myc ratio. We still detected strong EGFP reporter gene expression even if we located the optic fiber 0.8 mm above the virus injection site. These results suggest that the ST-Cal-Light is sufficient to label neurons even with the small intensity of blue light. This result is added in the Supplementary Figure. 3.

We also tested the reporter gene expression level depending on the blue light intensity in the previous study. We found that the level of gene expression increased as the intensity increased, but if the light intensity reached to some level, the level of gene expression became

no longer distinguishable (see *Supplementary Figure 3 from Lee et al, 2017*).

4) The authors call four labeling sessions (FR 8-12), with 550 seconds of blue light “weak labeling.” Did the authors try to label neurons after a single session? Similarly, foot shock experiments classically involve giving a single shock, and mice freeze on subsequent exposure. Tonegawa’s group has used c-fos driven tTA-expression (developed by Mayford)

mice to trace the memory engram of a single shock and manipulate it. The authors need to specify whether a single shock was sufficient to label neurons or not.

Thank the reviewer for these comments. As the reviewer pointed out, we performed “single session” labeling. While animals were performing a lever pressing task during the FR12 session (correspond to the training Day 10 in Fig. 2), we shone blue light. With the single session labeling, we observed a significant EGFP expression in a subset of neurons, indicating neuronal labeling during the single session is sufficient. The results are added in the Fig. 2f.

When we labeled hippocampal neurons by combining blue light and single foot shock, we detected some neurons expressing NpHR but inhibiting their activity did not significantly reduce the freezing behavior. This result is now added in the Supplementary Fig. 4.

5) Freeze experiments, please demonstrate that activating the Cal-light labeled neurons causes freezing.

To test if reactivation of the Cal-Light labeled neurons causes freezing behavior, we performed the same fear conditioning experiment where ChrimsonR was expressed instead of NpHR (Fig. 3f). Blue light was shone for 5 seconds concomitantly with the foot shock and this labeling was repeated 3 times. During the retrieval period, when mice are exposed to the conditioning context (context A), mice normally displayed freezing behavior (Fig. 3g). If animals were exposed to the different context (context B), in the control mouse group, freezing behavior was not observed. However, when neurons were labeled by ChrimsonR reporter, reactivation of their activity by 589 nm light triggered freezing behavior significantly higher (Fig. 3g).

6) Kainate acid –seizure experiments, seizures are measured best with a combination of EEG and behavior. Behavioral data alone is insufficient to support the authors’ claim of seizure suppression.

We thank the reviewer for this suggestion. We now performed the ST-Cal-Light experiments with EEG recording. An electrode for EEG recording was implanted over the hippocampus together with optic fiber. After cell labeling, the second KA injection caused ictal-like electrical discharges in the hippocampus, which was recognized by large magnitudes of EEG signals (Fig. 4j). These large amplitudes were more often detected in mouse group in the absence of yellow light. Shortly after KA injection, seizure activity was more frequently detected over time, and similar progressive seizure symptoms were also observed (Fig. 4k). When 589 nm light was shone, the average seizure activity progression as well as seizure score were suppressed, verifying again that the labeled neurons are directly implicated in seizure brain activity (Fig. 4k-4n).

7) The manuscript should be carefully edited to improve readability.

We thank the reviewer for this comment. In the revised manuscript, we added many new data and further clarification. We also edited the manuscript to improve readability.

Reviewers' Comments:

Reviewer #1:

Remarks to the Author:

The authors have made an excellent effort at addressing all our points. This manuscript is very well written and clear, and contributes a very important and powerful tool to the neuroscience community that I am sure will be of tremendous use to many.

We suggest a few very minor final edits:

1. Line 376, "could not make gene expression by itself", "Induce" or "activate" gene expression would be better.
2. I believe the authors define the wavelength of the light everywhere (e.g., line 169, "589 nm") _except_ for when it is blue. It would be useful to define the blue wavelength on the first instance in the Results section, e.g. on line 122. (Although "blue" is mentioned in the Introduction, I think the precise wavelength is less important in the Introduction.)
3. Line 381, I do not think the quotation marks around "mostly" are needed.

Reviewer #2:

Remarks to the Author:

The revised manuscript has addressed most of my previous concerns. This improved ST-Cal-Light method should be more useful for the neuroscience community. I have only a few minor points:

1. Line 134-135

There is a mismatch in the stimulation paradigm between the Main text and Methods.

In the main text: "a train of action potentials (10 pulses at 20 Hz per minute) was delivered for 15 minutes"

In the Methods: "One train (5 pulses at 10 Hz) of electrical stimulation was delivered per min with 10 sec-long blue light".

2. Fig. 1h, k

Y axis should be "Green Fluorescence (A.U.)". (A.U.) is missing.

Also, please describe in Methods how fluorescence is measured and normalized between samples.

3. Supplementary Fig. 3

In the panels, Myc is expressed in everywhere. However, GFP/Myc ratio on the figure reaches up to 5. How could this be possible?

Reviewer #3:

Remarks to the Author:

The authors have addressed my concerns by performing additional experiments.

REVIEWERS' COMMENTS

We wholeheartedly appreciate all reviewers again. Their valuable comments and suggestions were greatly improved the quality of manuscript. The remaining changes were marked as blue color text in the revised manuscript.

Reviewer #1 (Remarks to the Author):

The authors have made an excellent effort at addressing all our points. This manuscript is very well written and clear, and contributes a very important and powerful tool to the neuroscience community that I am sure will be of tremendous use to many.

We suggest a few very minor final edits:

1. Line 376, "could not make gene expression by itself", "Induce" or "activate" gene expression would be better.

We changed to "induce".

2. I believe the authors define the wavelength of the light everywhere (e.g., line 169, "589 nm") _except_ for when it is blue. It would be useful to define the blue wavelength on the first instance in the Results section, e.g. on line 122. (Although "blue" is mentioned in the Introduction, I think the precise wavelength is less important in the Introduction.)

We mentioned the precise wavelength for blue light, which is 473 nm.

3. Line 381, I do not think the quotation marks around "mostly" are needed.

Thank for the comment. We deleted the quotation marks.

Reviewer #2 (Remarks to the Author):

The revised manuscript has addressed most of my previous concerns. This improved ST-Cal-Light method should be more useful for the neuroscience community. I have only a few minor points:

1. Line 134-135

There is a mismatch in the stimulation paradigm between the Main text and Methods. In the main text: "a train of action potentials (10 pulses at 20 Hz per minute) was delivered for 15

minutes”

In the Methods: “One train (5 pulses at 10 Hz) of electrical stimulation was delivered per min with 10 sec-long blue light”.

We thank the reviewer for pointing out this mistake. Description in the method is a correct protocol. We changed the main text accordingly “a train of action potentials (5 pulses at 10 Hz per minute) with 10-sec long blue light was delivered for 15 minutes.”

2. Fig. 1h, k

Y axis should be “Green Fluorescence (A.U.)”. (A.U.) is missing.

Also, please describe in Methods how fluorescence is measured and normalized between samples.

We added (A.U.) in Fig. 1.

For G/R analysis, we included all neurons at each experimental condition and always green and red fluorescence level were analyzed in the consistent manner without adjusting background intensity. Red signals were similar across all conditions. This part is added in the Method section.

3. Supplementary Fig. 3

In the panels, Myc is expressed in everywhere. However, GFP/Myc ratio on the figure reaches up to 5. How could this be possible?

Because GFP signals represent quantitative measurement of gene expression, the GFP/Myc ratio was analyzed by the fluorescence intensity of GFP and Myc signals.

Reviewer #3 (Remarks to the Author):

The authors have addressed my concerns by performing additional experiments.